# SARS-CoV-2 host-shutoff impacts innate NK cell functions, but antibody-dependent NK activity is strongly activated through non-spike antibodies

Ceri Alan Fielding[1], Pragati Sabberwal[1], James C Williamson[2], Edward JD Greenwood[2], Thomas WM Crozier[2], Wioleta Zelek[1], Jeffrey Seow[3], Carl Graham[3], Isabella Huettner[3], Jonathan D Edgeworth[3,4], David A Price[1], Paul B Morgan[1], Kristin Ladell[1], Matthias Eberl[1], Ian R Humphreys[1], Blair Merrick[3,4], Katie Doores[3], Sam J Wilson[5], Paul J Lehner[2], Eddie CY Wang[1], Richard J Stanton[1]*

[1]Division of Infection and Immunity, School of Medicine, Cardiff University, Cardiff, United Kingdom; [2]Cambridge Institute for Therapeutic Immunology & Infectious Disease, Jeffrey Cheah Biomedical Centre, Cambridge Biomedical Campus, University of Cambridge, Cambridge, United Kingdom; [3]Department of Infectious Diseases, School of Immunology & Microbial Sciences, King's College London, London, United Kingdom; [4]Department of Infectious Diseases, Guy's and St Thomas' NHS Foundation Trust, London, United Kingdom; [5]MRC - University of Glasgow Centre for Virus Research, Glasgow, United Kingdom

*For correspondence:
stantonrj@cardiff.ac.uk

Competing interest: The authors declare that no competing interests exist.

**Abstract** The outcome of infection is dependent on the ability of viruses to manipulate the infected cell to evade immunity, and the ability of the immune response to overcome this evasion. Understanding this process is key to understanding pathogenesis, genetic risk factors, and both natural and vaccine-induced immunity. SARS-CoV-2 antagonises the innate interferon response, but whether it manipulates innate cellular immunity is unclear. An unbiased proteomic analysis determined how cell surface protein expression is altered on SARS-CoV-2-infected lung epithelial cells, showing downregulation of activating NK ligands B7-H6, MICA, ULBP2, and Nectin1, with minimal effects on MHC-I. This occurred at the level of protein synthesis, could be mediated by Nsp1 and Nsp14, and correlated with a reduction in NK cell activation. This identifies a novel mechanism by which SARS-CoV-2 host-shutoff antagonises innate immunity. Later in the disease process, strong antibody-dependent NK cell activation (ADNKA) developed. These responses were sustained for at least 6 months in most patients, and led to high levels of pro-inflammatory cytokine production. Depletion of spike-specific antibodies confirmed their dominant role in neutralisation, but these antibodies played only a minor role in ADNKA compared to antibodies to other proteins, including ORF3a, Membrane, and Nucleocapsid. In contrast, ADNKA induced following vaccination was focussed solely on spike, was weaker than ADNKA following natural infection, and was not boosted by the second dose. These insights have important implications for understanding disease progression, vaccine efficacy, and vaccine design.

## Editor's evaluation

By using a systematic proteomics approach, the authors demonstrated that SARS-CoV2 remodels the plasma membrane of infected human epithelial cells. Although the study indicates the

manipulation of different immune response pathways, it seems that in the focus of viral immunoevasion are natural killer (NK) cells, which play a crucial role in controlling early viral infection. However, antibody-dependent NK cell activation was observed later in the disease process. These findings could have implications for the understanding of SARS-CoV-2 control by the immune system and vaccine development.

## Introduction

COVID-19 studies have focused on interferon responses during the early innate phase of infection, and neutralising antibodies and virus-specific T-cells during the adaptive phase. In contrast, and despite their importance in antiviral protection (*Pierce et al., 2020*), considerably less is known about the role of Natural Killer (NK) cells in infection. NK cells bridge the innate and adaptive responses, and individuals with NK cell deficiencies suffer severe viral infections (*Orange, 2013*). NK cells respond rapidly to viruses, directly killing infected cells by releasing cytotoxic granules. They promote inflammation through the release of TNFα and IFNγ, and influence the induction of both B- and T-cell responses. Once virus-specific antibodies are induced, antibody-dependent NK activation (ADNKA) is generated through antibody linking cell-surface viral antigens with NK cell Fc receptors and subsequent antibody-dependent cellular cytotoxicity (ADCC).

The majority of studies on NK cells have focussed on their adaptive roles in established COVID-19 infection, with highly activated NK cells seen in both the periphery and the lung (*Maucourant et al., 2020*). The genetic loss of NKG2C (a hallmark of adaptive NK cells, capable of enhanced ADCC) is also a risk factor for severe disease (*Vietzen et al., 2021b*), implying a protective role in COVID-19 infection. In support of this, in animal models, Fc-mediated functions correlate with protection in vaccination (*Gorman et al., 2021*), and monoclonal antibody-mediated control of SARS-CoV-2 infection is enhanced by Fc-dependent mechanisms (*Chan et al., 2020*; *Schäfer et al., 2021*; *Winkler et al., 2020*; *Suryadevara et al., 2021*; *Winkler et al., 2021*; *Yamin et al., 2021*), which enable control of disease even in the absence of neutralising activity (*Beaudoin-Bussières et al., 2021*). Furthermore, although neutralising monoclonal antibodies are effective when administered prophylactically, Fc-activity is required for efficacy when administered to animals therapeutically (*Yamin et al., 2021*).

These animal studies demonstrate potentially important roles for adaptive NK cell responses in COVID-19. Despite this, our understanding of ADCC following infection and vaccination in humans remains limited. Studies to date have assumed spike protein is the dominant ADCC target, and have tested plate-bound protein or transfected cells (*Barrett et al., 2021*; *Herman et al., 2021*; *Zohar et al., 2020*; *Tauzin et al., 2021*; *Anand et al., 2021*; *Lee et al., 2020*; *Dufloo et al., 2020*; *Tortorici et al., 2020*; *McCallum et al., 2021*; *Cathcart et al., 2021*). Using these systems, they have shown that ADNKA-inducing antibodies are generated following SARS-CoV-2 infection (*Herman et al., 2021*; *Anand et al., 2021*; *Lee et al., 2020*; *Dufloo et al., 2020*), are induced by vaccination in humans (*Barrett et al., 2021*; *Tauzin et al., 2021*), and can be mediated by a subset of neutralising monoclonal antibodies targeting the spike protein (*Tortorici et al., 2020*; *McCallum et al., 2021*; *Cathcart et al., 2021*). However, these responses have not been tested against SARS-CoV-2 infected cells; the presence, abundance, accessibility, and conformation of viral glycoproteins can be very different during productive infection. Furthermore, although viral entry glycoproteins such as spike can be found on the infected cell surface, and can mediate ADCC, other viral proteins can be the dominant mediators of ADCC during infection; indeed, we have shown that in other viruses, non-structural proteins are often the dominant targets (*Vlahava et al., 2021*).

In addition, there is little information on the NK cell response to SARS-CoV-2 infected cells during the innate phase of infection, despite this process being important to virus pathogenesis; successful viruses must counteract robust NK activation to establish infection, and their ability to do this is critical to their ability to cause disease (*Patel et al., 2018*; *Berry et al., 2020*). Whether SARS-CoV-2's ability to infect humans is dependent on viral evasion of innate cellular immunity, is unknown. NK cell activation is complex and depends on the balance of ligands for a wide range of inhibitory or activating receptors (*Kumar, 2018*). These ligands can be induced on the surface of target cells in response to stress, infection, or transformation. For example, the stress ligands MICA, MICB, and ULBP1-6, all bind to the ubiquitously expressed activating receptor NKG2D. To limit NK cell activation, many viruses manipulate these NK activating ligands, reducing NK-cell-mediated control (*Patel*

*et al., 2018*). Whether similar manipulations underlie the ability of SARS-CoV-2 to cause disease in humans remains unclear.

To address these critical gaps in our understanding of the SARS-CoV-2-specific NK cell response, we used quantitative proteomics to determine how SARS-CoV-2 infection affected cell surface protein expression (*Rihn et al., 2021*). We found that SARS-CoV-2-infected cells downregulate multiple activating NK ligands, resulting in a reduced NK cell response to infected cells. This suggests a novel mechanism by which SARS-CoV-2 counteracts innate immunity early in infection, to establish disease. The development of humoral immunity following natural infection resulted in robust ADNKA against infected cells, and high levels of proinflammatory cytokines; these responses persisted for at least 6 months in most patients. Proteomics identified four cell-surface viral proteins capable of driving this response, leading to the surprising discovery that ADNKA is not primarily driven by spike antibody, but was dominated by antibody to other viral proteins such as nucleocapsid, membrane and ORF3a; this defines a novel role for antibodies targeting the immunodominant nucleocapsid protein. In agreement with these findings, and in contrast to plate-bound protein or transfected cells, spike was a poor activator of ADNKA following vaccination when tested using infected cells. This implies that the breadth and magnitude of ADNKA responses is limited in current vaccination strategies. The inclusion of antigens such as nucleocapsid would recruit additional effector mechanisms, and may help maintain vaccine efficacy following mutation of spike in naturally circulating virus variants.

## Results

### SARS-CoV-2 remodels the plasma membrane proteome

Immune cells recognise and interact with infected cells through cell surface ligands. We therefore wanted to gain a comprehensive and unbiased overview of how SARS-CoV-2 infection changes the plasma membrane protein landscape. We performed plasma membrane enrichment through selective aminooxy-biotinylation (Plasma Membrane Profiling; PMP), a methodology that we previously developed (*Weekes et al., 2010*; *Weekes et al., 2014*) and have validated extensively against human immunodeficiency virus (HIV) (*Matheson et al., 2015*), human cytomegalovirus (HCMV) (*Vlahava et al., 2021*; *Weekes et al., 2014*; *Fielding et al., 2017*; *Hsu et al., 2015*; *Weekes et al., 2013*), and herpes simplexvirus (HSV) (*Soh et al., 2020*) infection.

High infection rates are critical to minimise confounding effects from bystander cells. We therefore applied PMP to SARS-CoV-2 infected A549-ACE2-TMPRSS2 (AAT) cells, which are highly permissive to SARS-CoV-2 (*Rihn et al., 2021*) and infection at a high multiplicity of infection (MOI) results in over 90% infection by 24 hr (*Figure 1A*). We utilised TMT-based quantification to compare plasma membrane (PM) protein abundance in uninfected cells, and cells infected for 12, 24, and 48 hr (*Figure 1B*). In total, 4953 proteins were quantitated, including 2227 proteins with gene ontology annotation associated with the cell surface and plasma membrane (*Figure 1—figure supplement 1A*). Importantly however, the plasma membrane annotated proteins made up nearly 80% of the total protein abundance, indicating a high degree of enrichment for plasma membrane proteins, comparable with our previous PMP datasets (*Vlahava et al., 2021*; *Weekes et al., 2010*; *Weekes et al., 2014*; *Matheson et al., 2015*; *Fielding et al., 2017*; *Hsu et al., 2015*; *Weekes et al., 2013*; *Soh et al., 2020*). The detection of low abundance proteins that are not plasma-membrane annotated reflects a combination of incorrect annotations, and contaminating intracellular proteins that were incompletely removed following PMP.

Five SARS-CoV-2 viral proteins were detected (see later), including spike, ORF3A, and membrane, which have previously been annotated as having a cell surface localisation (*Figure 1C*). Consistent with prior reports, the viral receptor ACE2 was strongly downregulated after infection (*Patra et al., 2020*), while TMPRSS2 expression was unaffected (*Figure 1C*). Overall, of the 2227 PM proteins quantified with more than one peptide, 914 showed changes which were both significant and with a greater than 1.5-fold alteration over the course of the experiment. Clustering of proteins by their temporal pattern of change has previously shown utility in defining classes of regulated proteins and predicting underlying mechanisms (*Matheson et al., 2015*; *Greenwood et al., 2016*). Here, we used K-means clustering to define five groups of significantly altered proteins by their temporal profile.

To assess the innate immune response triggered by SARS-CoV-2 infection, the clusters were interrogated for the presence of a predefined list of interferon alpha-inducible genes. Thirteen of the

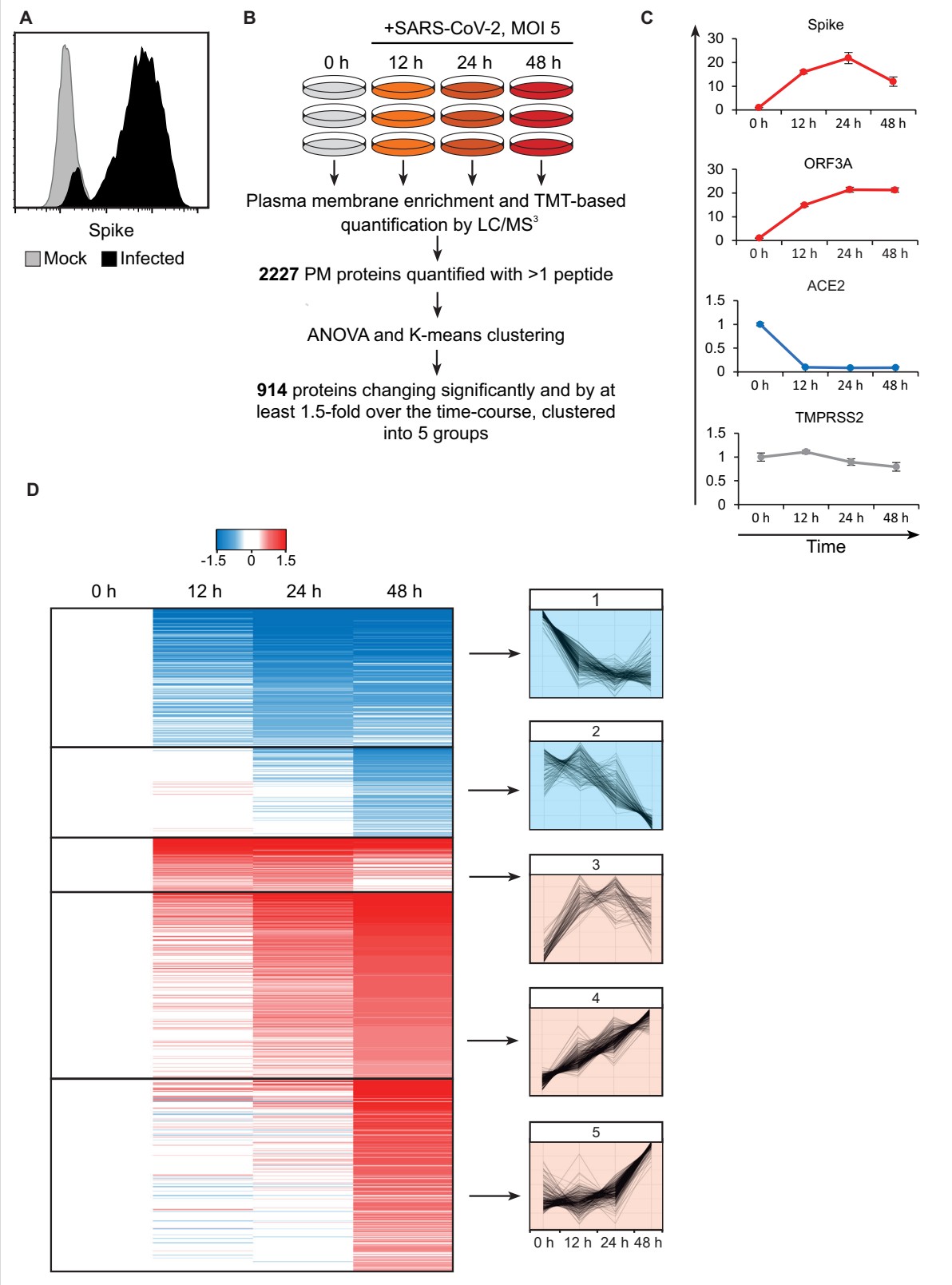

**Figure 1.** SARS-CoV-2 remodels the plasma membrane proteome. (**A**) AAT cells were infected with SARS-CoV-2 in biological triplicate (MOI = 5). Twenty-four hr later, they were detached with trypsin, fixed and permeabilised, stained for Spike protein, and analysed by flow cytometry. (**B**) Schematic of plasma membrane profiling and analysis pathway. (**C**). Examples of temporal profiles of viral and cellular genes, fold change is compared to 0 hr

*Figure 1 continued on next page*

*Figure 1 continued*

timepoint. Data points show mean ± SD. (**D**). Left – heat map of the 914 significantly changing proteins clustered by k-mean, colour indicates log2 fold change compared to 0 hr, right, Z-score normalised temporal profiles of proteins within each cluster.

The online version of this article includes the following figure supplement(s) for figure 1:

**Figure supplement 1.** Plasma membrane proteins are enriched in the PMP dataset.

**Figure supplement 2.** MS method parameters.

**Figure supplement 3.** Search parameters for MS data processing.

23 detected PM genes associated with the interferon response fell into cluster 5 (*Figure 1—figure supplement 1B*, *Supplementary file 2*), which describes proteins with an increase in expression at 48 hr, examples shown in *Figure 1—figure supplement 1C*. As the SARS-CoV-2 viral replication cycle is completed in less than 24 hr (*Lei et al., 2020*), SARS-CoV-2 is able to evade innate immune recognition and/or suppress IFN signalling until long after virus egress has been achieved.

To identify potential innate and adaptive immune evasion strategies employed by SARS-CoV-2, we focused on cluster 1, which describes proteins downregulated from early in the time course. This cluster was subject to gene ontology and pathway analysis to define proteins related by functional and structural similarities (*Figure 2*). Several classifications enriched in this cluster are particularly relevant, including the downregulation of several cytokine receptors, including IFNAR1. Much of the SARS-CoV-2 genome is dedicated to antagonising a type I interferon response within an infected cell (*Lei et al., 2020*) and the plasma membrane downregulation of IFNAR1 presents a previously undescribed strategy to prevent signalling from exogenous IFN.

Also of note is the downregulation of heparan sulfate proteoglycans, the syndecan (SDC), glypican (GPC), and neurophillin (NRP) proteins, which together encompass the families of heparan sulfate modified proteins expressed on epithelial cells (*Sarrazin et al., 2011*). NRP1 is a cell entry factor for SARS-CoV-2 (*Cantuti-Castelvetri et al., 2020*; *Daly et al., 2020*), and SARS-CoV-2 spike is also reported to bind heparan sulphated proteins (*Clausen et al., 2020*). The decrease in cell surface expression of heparan sulfated proteins may therefore enable the egress of viral particles and prevent super-infection, and is reminiscent of the capacity of other viruses to downregulate both primary viral receptors and co-receptors (*Landi et al., 2011*).

## SARS-CoV-2 inhibits the synthesis of multiple NK cell ligands and reduces NK activation

Our experience with PMP has emphasised the modulation of NK ligands as an important immune evasion mechanism (*Fielding et al., 2017*). As these ligands are poorly annotated in public databases, we investigated the PMP dataset for previously described ligands (*Kumar, 2018*), of which 18 were detected (*Figure 2—figure supplement 1*). MHC-I proteins HLA-B to HLA-E were upregulated in cluster 5, likely reflecting a late IFN-mediated upregulation, while HLA-A was unchanged. By contrast, the activating NK-cell ligands MICA, ULBP2, B7-H6 (NCR3LG1), and Nectin1 (which is inhibitory in mice, but recently reported as activating in humans *Holmes, 2019*) were significantly downregulated in cluster 1. This was confirmed by flow cytometry in multiple cell types (*Figure 3A*, *Figure 3—figure supplement 1A*). These responses were dependent on virus replication, because no alterations in ligand abundance were observed when virus was heat-inactivated (*Figure 3—figure supplement 2A*).

Many of the best characterised mechanisms by which viruses inhibit the function of NK ligands involve inhibition of cell-surface transport and/or proteasomal or lysosomal degradation of the mature protein (*Patel et al., 2018*; *Berry et al., 2020*). To determine whether SARS-CoV-2 targeted NK ligands in the same way we measured their abundance, and their sensitivity to digestion with EndoH, which cleaves N-linked glycans from proteins that have not yet transited (or have been retained in) the endoplasmic reticulum (ER). Digestion with PNGaseF (which cleaves N-linked glycans irrespective of subcellular trafficking) served as a control to show the effect of complete deglycosylation. We were unable to detect endogenous NK ligands by western blot (data not shown), and therefore overexpressed MICA, ULBP2, and B7-H6 using replication deficient adenovirus vectors (RAd). Following infection with SARS-CoV-2, levels of all three NK ligands were substantially reduced, however there was no alteration in the pattern of EndoH sensitivity (*Figure 3B*). Thus, SARS-CoV-2 infection does not result in intracellular retention of NK ligands, but instead reduces the overall abundance of protein.

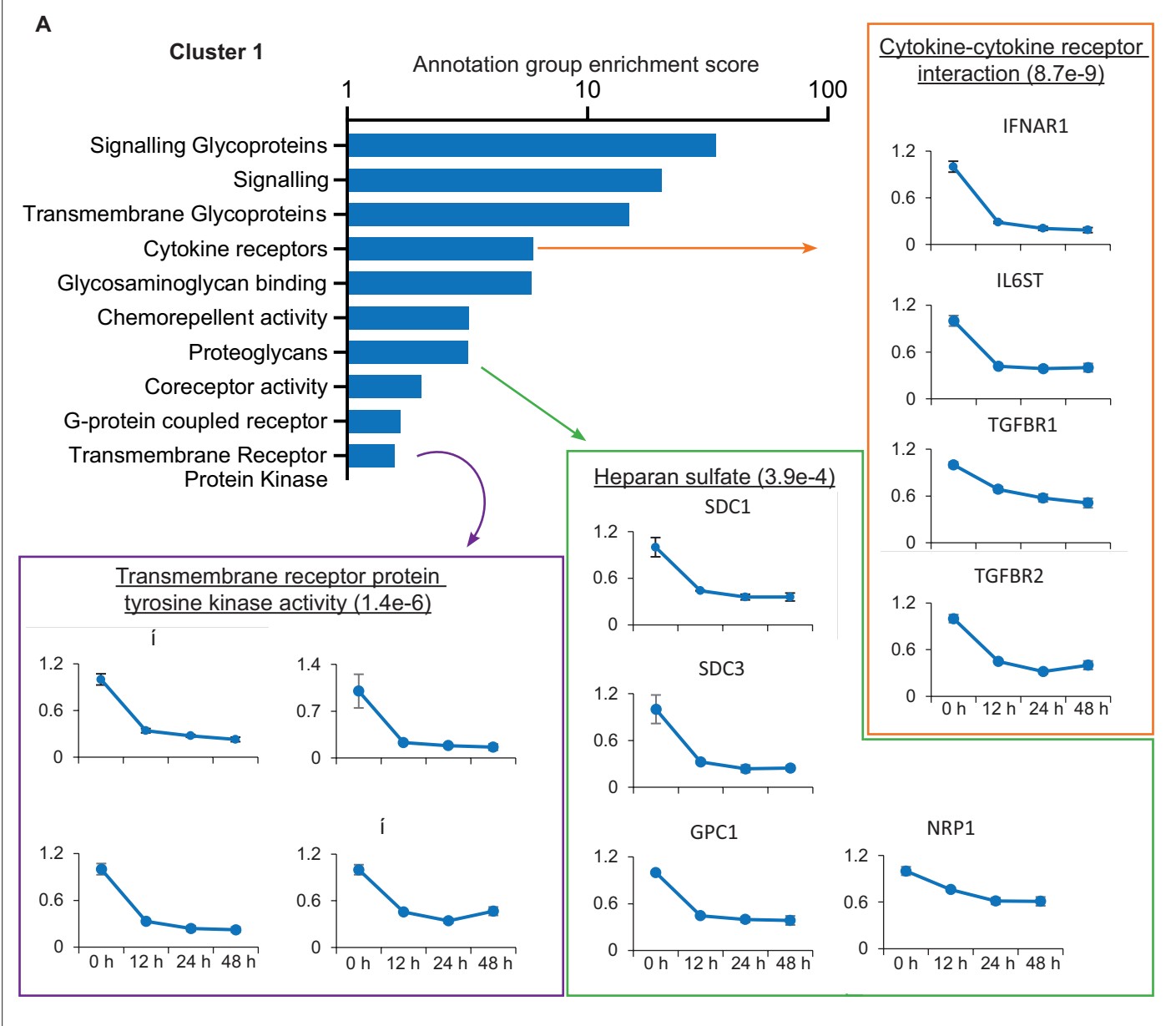

**Figure 2.** SARS-CoV-2 manipulates multiple classes of plasma membrane proteins. Enriched gene ontology and pathway annotations of proteins downregulated in temporal cluster 1 were condensed to groups of related terms, bar chart shows the enrichment score of these groups. Examples of annotation terms falling within each group are shown underlined with Bejamani-Hochberg adjusted p-values in parentheses, alongside four examples of proteins for each annotation.

The online version of this article includes the following figure supplement(s) for figure 2:

**Figure supplement 1.** NK ligands detected in the PMP dataset.

To determine whether this occurred by inhibition of synthesis or degradation of mature protein, we treated cells with proteasomal (MG132) or lysosomal (Leupeptin) inhibitors after infection (*Figure 3C*). When tested with overexpressed MICA or B7-H6 (ULBP2 was not tested), MG132 did not result in recovery of NK ligand levels. Leupeptin resulted in a small recovery of B7-H6; however, this was seen in both Mock and infected cells, implying that it reflects constitutive turnover of the ligand rather than virus-specific targeting, and recovery in infected cells remained reduced compared to untreated mock infected cells. Overall, these results are consistent with the virus targeting NK ligands at the level of synthesis, rather than post-translationally. We therefore investigated whether NK ligands were

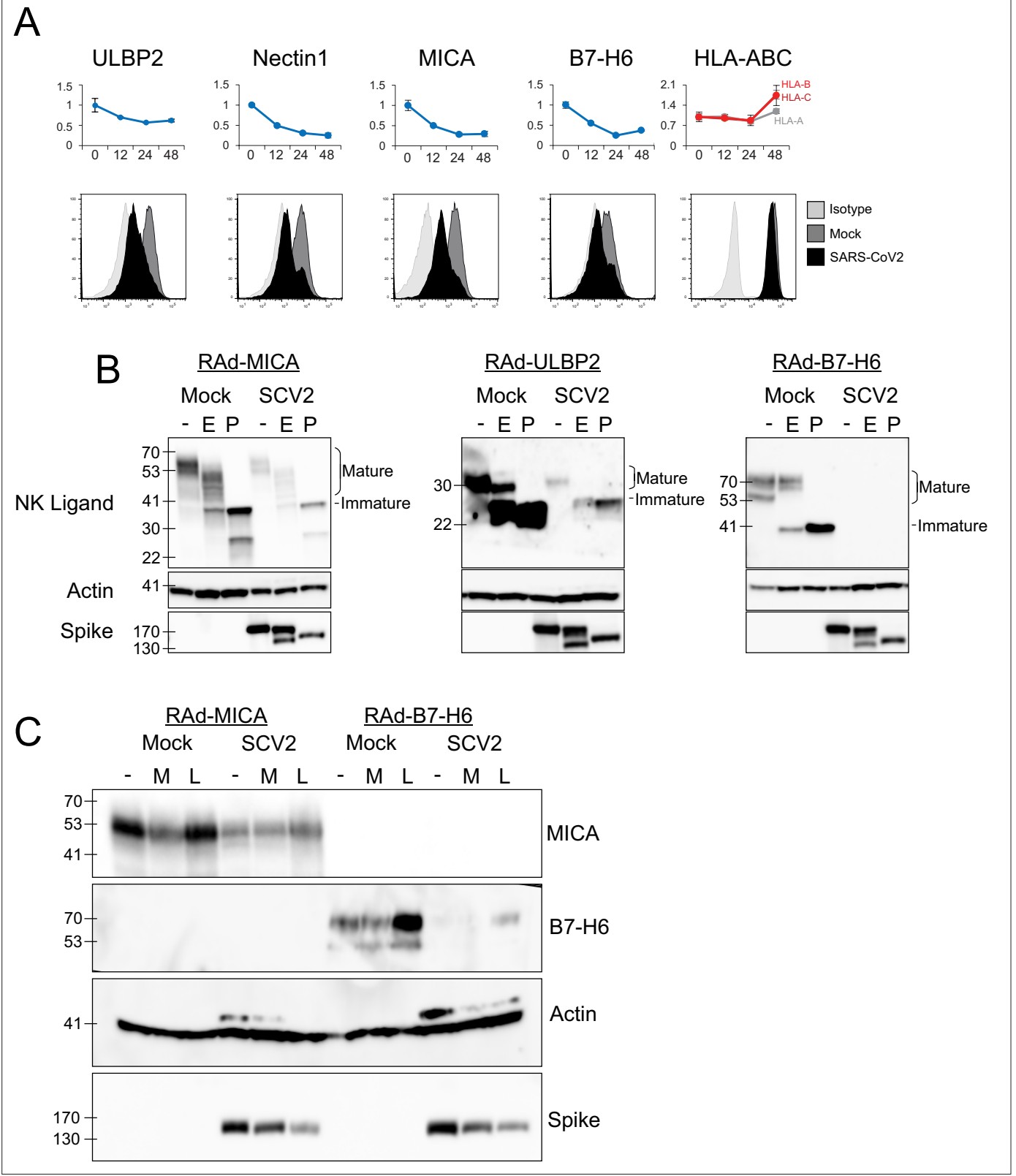

**Figure 3.** SARS-CoV-2 infection inhibits the expression of multiple NK ligands. (**A**) AAT cells were either mock infected, or infected with SARS-CoV-2 for 24 hr (MOI = 5), detached using TrypLE, then stained for the indicated NK cell receptors before being analysed by flow cytometry (bottom). Plots for the same proteins from PMP are included for reference (top) (**B**) AAT cells were infected with RAd-MICA, RAd-ULBP2, or RAd-B7-H6, then after 48 hr either mock infected, or infected with SARS-CoV-2 (SCV2) for a further 24 hr (MOI = 5). Lysates were either kept undigested, or digested with EndoH (**E**), or

*Figure 3 continued on next page*

*Figure 3 continued*

PNGaseF (**P**), then western blotted for the indicated proteins. Mature cell surface glycoproteins are resistant to EndoH, while immature (ER resident) forms are digested and therefore run with a smaller mass following EndoH digestion (**C**) AAT cells were infected with RAd-MICA or RAd-B7-H6, then after 24 hr either mock infected, or infected with SARS-CoV-2 (SCV2), and treated with MG132 (**M**) or Leupeptin (**L**).Twenty-four hour later lysates were made, and analysed by western blot for the indicated proteins. (Note: The additional band seen when staining for actin in the presence of SARS-CoV-2 may be non-specific).

The online version of this article includes the following figure supplement(s) for figure 3:

**Figure supplement 1.** SARS-CoV-2 infection downregulates multiple NK ligands in CACO2 cells.

**Figure supplement 2.** Modulation of NK activity is dependent on virus replication.

particularly sensitive to inhibition of synthesis, by treating cells expressing MICA, B7-H6, or a control protein (GFP) with the translation inhibitor cycloheximide (ULBP2 was not tested). This resulted in rapid loss of MICA, ULBP2, and B7-H6, but not GFP (*Figure 4A*). Finally, to determine which viral proteins were responsible for these effects, we transfected cells with a library of previously validated plasmids expressing all 29 SARS-CoV-2 ORFs individually (*Gordon et al., 2020*), and assessed them for levels of NK ligands on the cell surface. Only two SARS-CoV-2 ORFs resulted in a reduction of staining; Nsp1 and Nsp14 (*Figure 4B*). For MICA and ULBP2, Nsp14 consistently resulted in the strongest reduction, with Nsp1 being weaker. For Nectin1 and B7-H6, effects were weaker, but both Nsp1 and Nsp14 resulted in similar reductions.

Both Nsp1 and Nsp14 are involved in the ability of SARS-CoV-2 to degrade host RNA (Nsp1), and inhibit host protein translation (Nsp1 and Nsp14), and these activities have been implicated in their ability to counteract the interferon response within the infected cell (*Yuan et al., 2021*; *Hsu et al., 2021*). Together with our biochemical analysis, this data suggests that NK ligands are highly susceptible to inhibition of synthesis, and that SARS-CoV-2 manipulation of host protein production could impact the NK response in addition to the interferon response. To determine whether this was the case, we incubated SARS-CoV-2 infected cells with PBMC, and measured (i) degranulation of CD3$^-$CD56$^+$ NK cells (*Figure 4C*; *Figure 3—figure supplement 2B*), (ii) production of IFNγ (*Figure 4D*), and (iii) TNFα (*Figure 4E*, *Figure 3—figure supplement 2B*). Consistent with data using other viruses *Vlahava et al., 2021*, CD107a was recirculated on a higher proportion of NK cells than IFNγ and TNFα. Nevertheless, all three markers of NK activation were significantly reduced in SARS-CoV-2 infected versus mock infected cells, suggesting that the virus has evolved to inhibit NK cell responses. Comparable results were seen at both 24 hr and 48 hr post-infection (data not shown).

## NK-cell evasion is overcome by antibody-dependent activation

In addition to their role in innate immunity through interactions with activating and inhibitory NK receptors early in infection, NK cells play additional roles following the development of adaptive immunity via ADNKA/ADCC, in which CD16 interacts with the Fc portion of antibodies bound to targets on the surface of infected cells. To analyse this aspect of NK function, PBMC were incubated with infected cells in the presence of sera from individuals that were seronegative or seropositive for SARS-CoV-2. Certain NK cell subsets degranulate more effectively in response to antibody bound targets. These NK cells can be differentiated by the presence of NKG2C and CD57 on the cell surface (*Stary and Stary, 2020*). However, NKG2C is only found in a subset of donors who are HCMV seropositive, and a proportion of these individuals are NKG2C$^{null}$ due to a deletion in the KLRC2 gene. As a result, NKG2C cannot be used as a marker in all donors. We have previously shown that CD57 is sufficient to identify higher responding NK cells in all donors, and activation of this population correlates with the ability of antibodies to mediate ADCC (*Fielding et al., 2021*; *Wang et al., 2018*). We therefore focussed on this population here; identical patterns of activation were observed in the CD57-negative population, albeit of slightly smaller magnitude (not shown). Non-specific activation was controlled for by testing against mock-infected cells throughout.

Degranulation (*Figure 5A*), as well as production of IFNγ (*Figure 5B*) and TNFα (*Figure 5C*), were all significantly increased in the presence of serum containing anti-SARS-CoV-2 antibodies, across a range of different NK donors (*Figure 5D–F*). ADNKA was dependent on virus replication, since heat inactivated virus did not lead to serum-dependent NK degranulation (*Figure 5—figure supplement 1A*). NK cells were activated equivalently whether purified NK cells or PBMC were used (*Figure 5—figure supplement 1B*), thus these responses are due to direct stimulation of NK cells, rather than

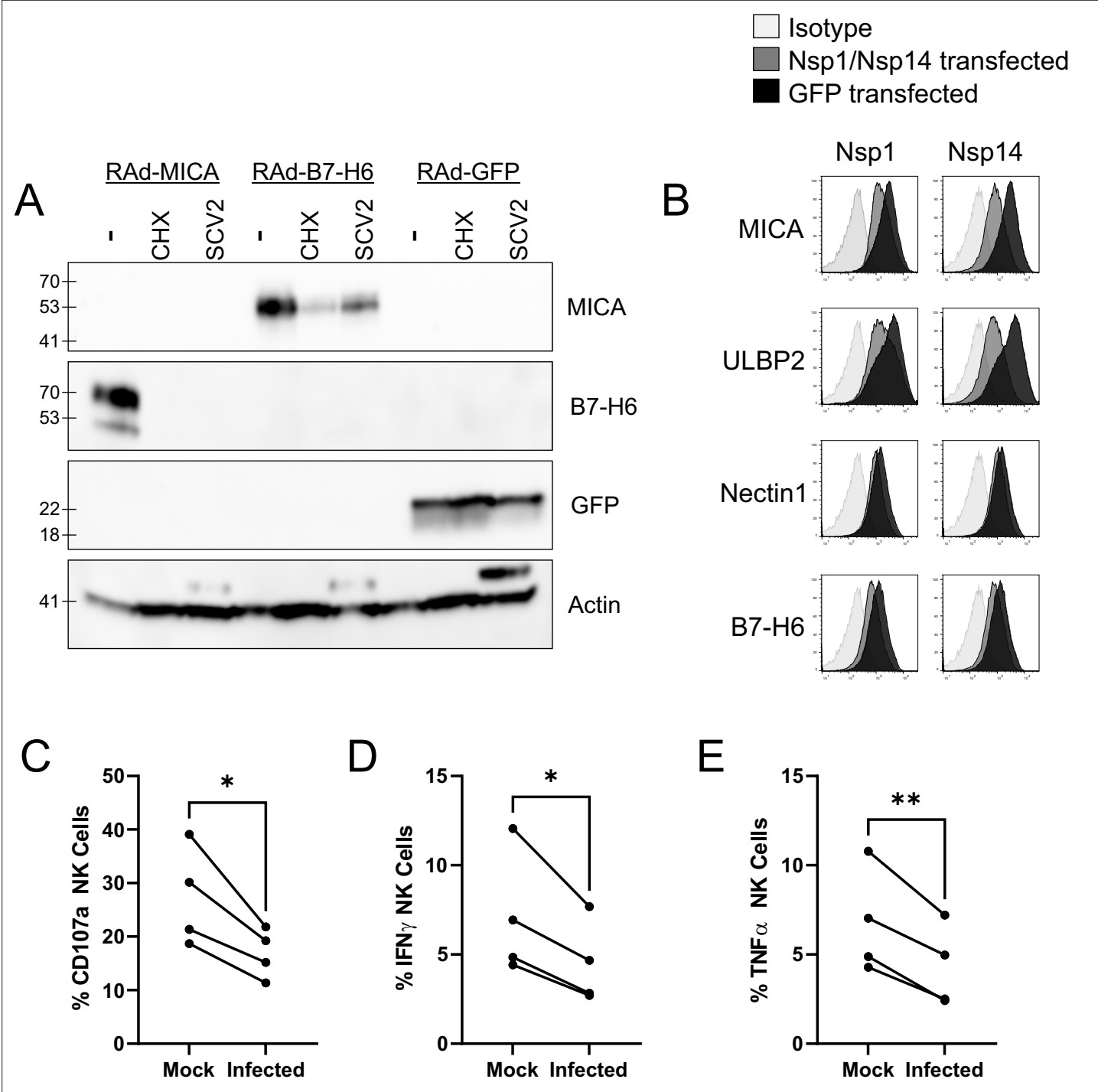

**Figure 4.** SARS-CoV-2 infection restricts the expression of multiple NK ligands, inhibiting NK activation. (**A**) AAT cells were infected with RAd-MICA, RAd-B7-H6, or RAd-GFP, then after 24 hr either mock infected, infected with SARS-CoV-2 (SCV2), or treated with Cycloheximide (CHX). Twenty-four hr later, cells were lysed and analysed by western blot for the indicated proteins (Note: The additional band seen when staining for actin in the presence of SARS-CoV-2 may be non-specific). (**B**) 293T cells were transfected with expression plasmids for Nsp1 or Nsp14, or GFP as a control. Twenty-four hr later, cells were detached with HyQtase and stained for the indicated proteins. (**C–E**) AAT cells were either mock infected, or infected with SARS-CoV-2 for 24 h (MOI = 5), detached with TrypLE and mixed with interferon stimulated PBMC for 5 hr in the presence of golgistop, golgiplug, and CD107a antibody, before staining for CD3/CD56 and Live/Dead Aqua. Cells were then fixed, permeabilised, and stained for TNFα and IFNγ. Cells were gated on live NK cells, and the percentage of cells positive for CD107a, TNFα, and IFNγ calculated. Individual assays were run in technical triplicate, with data shown from four assays using different donors. Kruskal–Wallis, *p<0.05, **p<0.01.

The online version of this article includes the following figure supplement(s) for figure 4:

*Figure 4 continued on next page*

*Figure 4 continued*

**Figure supplement 1.** Example gating strategy for NK assays.

other cells responding to antibodies and releasing NK stimulating cytokines. Viral immune evasion can therefore be overcome through antibody dependent mechanisms, suggesting that in addition to neutralising virus, SARS-CoV-2 antibodies may aid virus clearance through ADCC.

The magnitude of the neutralising antibody response to SARS-CoV-2 correlates with severity of disease; individuals with more severe disease generate higher levels of neutralising antibody (*Seow et al., 2020*; *Dan et al., 2021*; *Wajnberg et al., 2020*). Although memory B-cell responses persist, levels of circulating antibody can decrease over time following resolution of disease, such that in some mildly infected people they become undetectable after a few months (*Seow et al., 2020*; *Dan et al., 2021*; *Wajnberg et al., 2020*). To determine whether the same was true of ADCC responses, we assessed ADNKA in a cohort of donors with known disease status, for whom longitudinal sera were available (*Figure 5G*). ADNKA responses were generally of greater magnitude in individuals with more severe disease, with responses remaining detectable during the 150–200 day follow-up. Amongst those with milder disease, strong initial responses were maintained over time, while weaker responses decreased, becoming virtually undetectable by around 200 days. There was a moderate correlation between the ability of antibodies to neutralise SARS-CoV-2, and to activate ADNKA in response to SARS-CoV-2 (*Figure 5H*), with a $R^2$ value of 0.7. Thus, although the antibodies mediating these activities may be induced in a similar manner, the antibodies mediating the two responses may not be identical.

## Multiple SARS-CoV-2 ORFs mediate ADCC

There has been a major focus on spike as an activator of ADCC (*Barrett et al., 2021*; *Herman et al., 2021*; *Zohar et al., 2020*; *Tauzin et al., 2021*; *Anand et al., 2021*; *Lee et al., 2020*; *Dufloo et al., 2020*; *Tortorici et al., 2020*; *McCallum et al., 2021*; *Cathcart et al., 2021*). It is certainly a potential target during infection as, despite the fact that virion particles bud internally rather than from the cell membrane, both our PMP and flow cytometric analysis detected substantial levels of spike protein on the infected cell surface (*Figure 6A, C*, *Figure 5—figure supplement 1C, D*). However, our studies with other viruses have demonstrated that although viral proteins involved in virion binding and entry are found on the cell surface, and can be bound by antibody, they are not necessarily the strongest mediators of ADCC activity (*Vlahava et al., 2021*). PMP also detected membrane (M), nucleocapsid (N), ORF3a, and ORF1ab on the cell surface. ORF1ab is expressed as a polyprotein, which is proteolytically cleaved into multiple non-structural proteins (NSPs). Analysis of individual peptides representative of the proteolytically processed NSPs did not reveal any specific NSP enrichment. Furthermore, comparing the relative abundance of viral proteins in PMP, ORF1ab was only present at very low levels (1% of all viral proteins, significantly lower than any other SARS-CoV-2 protein). Taken together, it is unlikely that any NSPs are surface-exposed. Nevertheless, we performed bioinformatic analysis of all viral proteins for the presence of potential transmembrane domains or signal peptides, revealing five additional candidate cell-surface proteins (Nsp4, Nsp6, ORF6, ORF7a, ORF7b). We transfected cells with previously validated expression constructs for each of these genes (*Gordon et al., 2020*), and screened them for their ability to activate NK cells in an antibody-dependent manner. We used sera from patients with severe COVID-19 disease, that had previously generated the strongest ADNKA signals against infected cells (*Figure 6B*). As expected, spike primed a substantial ADNKA response. However, in support of our PMP data, expression of ORF3a, membrane, and nucleocapsid, were also capable of priming ADNKA in the presence of serum containing SARS-CoV-2 antibodies, while the Nsps did not.

Interestingly, nucleocapsid readily activated ADNKA despite the fact that this protein has previously been assumed to be intracellular. We have previously observed the exquisite sensitivity of ADNKA, which can be activated in response to levels of virus protein that are difficult to detect by flow cytometry (*Vlahava et al., 2021*). We have also shown that antigen levels can be very different in the context of virus infection, compared with transfection (*Vlahava et al., 2021*). In support of this, nucleocapsid levels were much higher on the surface of infected cells than transfected cells (*Figure 6C*); detection of nucleocapsid was dependent on virus replication (*Figure 5—figure supplement 1C*), and occurred following infection of multiple different cell types (*Figure 5—figure supplement 1D*). The

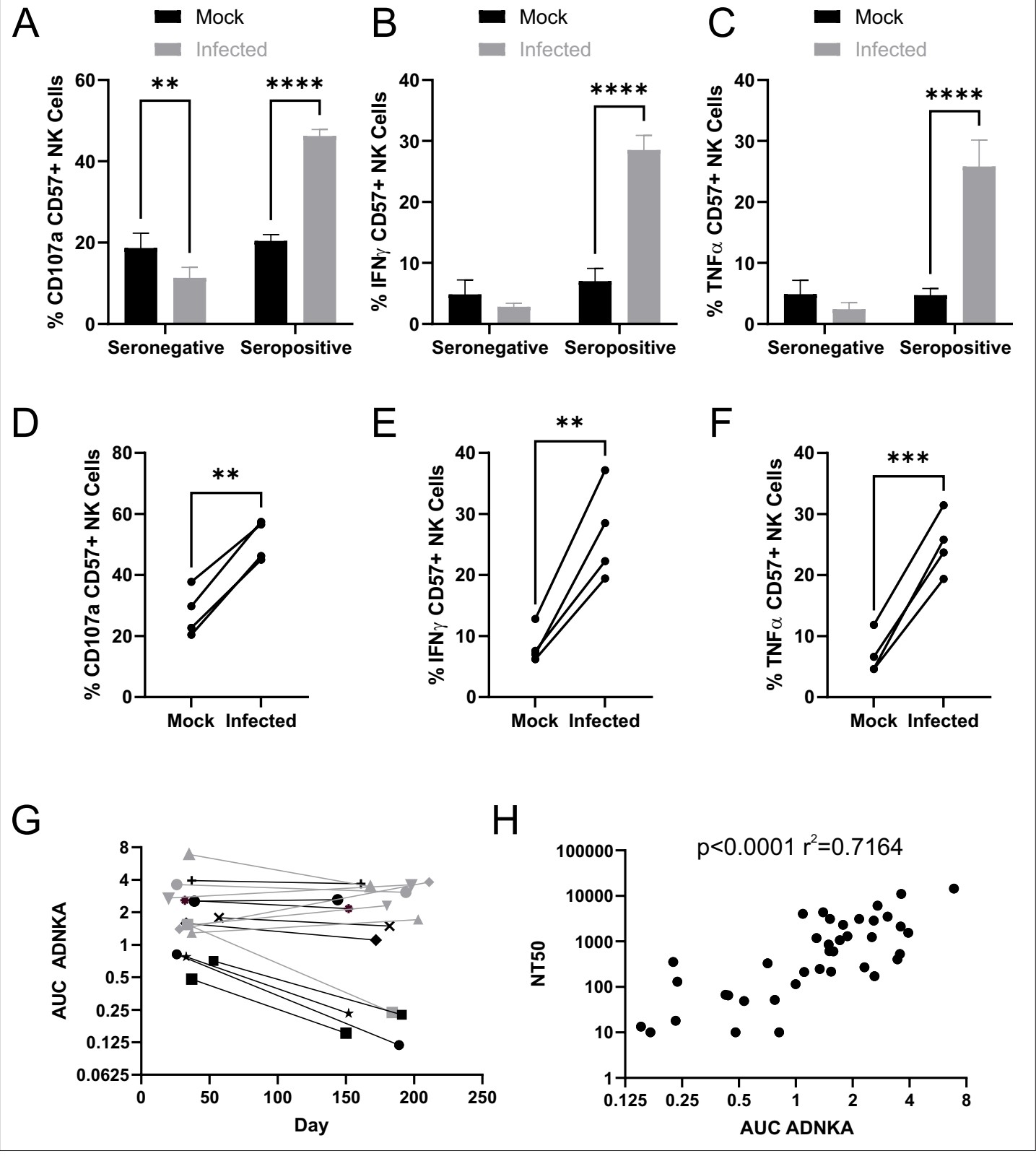

**Figure 5.** SARS-CoV-2 inhibition of NK activation can be overcome via ADNKA. AAT cells were either mock infected, or infected with SARS-CoV-2 for 24 hr (MOI = 5), detached using TrypLE, then mixed with PBMC in the presence of golgistop, CD107a antibody, and serum from donors who were seronegative or seropositive for SARS-CoV-2. After 5 hr, cells were stained for CD3, CD56, CD57, and live/dead aqua, fixed, permeabilised and stained for IFNγ and TNFα, then analysed by flow cytometry. (**A–F**) assays were performed using 1% serum, and the percentage of CD57 +NK cells positive for

*Figure 5 continued on next page*

*Figure 5 continued*

CD107a (**A, D**), TNFα (**B, E**), and IFNγ (**C, F**) were calculated. Individual donors were performed in technical triplicate (**A–C**), two-way ANOVA **$p<0.01$, ****$p<0.0001$. Assays from multiple donors were also compared (**D–F**), Kruskal–Wallis **$p<0.01$, ***$p<0.001$. (**G–H**) assays were performed using a serial threefold dilution of serum, starting from 3.3%, then the area under the curve (AUC) calculated. (**G**) Longitudinal serums from donors who experienced mild (black) or severe (grey) were used. (**H**) In addition to AUC for ADNKA activity, serums were tested for their ability to neutralise SARS-CoV-2 infection of VeroE6 cells, and the NT50 calculated, then compared to the AUC for ADNKA; Spearman rank correlation analysis is shown.

The online version of this article includes the following figure supplement(s) for figure 5:

**Figure supplement 1.** ADNKA is dependent on virus replication, and occurs through direct action on NK cells.

presence of nucleocapsid on the cell surface was also supported by immunofluorescence staining of non-permeabilised infected cells (*Figure 6D*), where distinct patches of nucleocapsid were observed that sometimes, but not always, coincided with cell-cell junctions. Given how well even very low levels of surface nucleocapsid activated ADNKA, this suggested that multiple proteins could be major contributors to ADCC during SARS-CoV-2 infection.

## Monoclonal anti-spike antibodies only weakly mediate ADNKA despite binding strongly to infected cells

To determine the relative contribution of spike, and of different antigenic sites on spike, to ADNKA during SARS-CoV-2 infection, we tested a panel of 26 human anti-spike monoclonal antibodies, which had been isolated from naturally infected donors, and cloned as IgG1 constructs (*Graham et al., 2021*). The panel included antibodies targeting the RBD, S1, NTD, and S2 domains, neutralising and non-neutralising, and those targeting multiple distinct epitopes (*Supplementary file 4*).

All 26 antibodies bound efficiently to infected cells, at levels comparable to polyclonal serum (*Figure 7—figure supplement 1*). However, only five of these antibodies were capable of triggering degranulation (*Figure 7A*), of which two triggered TNFα (*Figure 7—figure supplement 2A*), and one IFNγ (*Figure 7—figure supplement 2B*). Importantly, these NK responses were at dramatically lower levels than seen with polyclonal serum from a naturally infected donor, even when combined (*Figure 7B*).

A number of previous publications have used cells transfected with spike expression constructs to suggest that monoclonal anti-spike antibodies can induce potent ADNKA, which would appear to contradict our data (*Chan et al., 2020*; *Schäfer et al., 2021*; *Winkler et al., 2020*; *Suryadevara et al., 2021*; *Winkler et al., 2021*; *Yamin et al., 2021*). We therefore tested a subset of our monoclonal antibodies against transfected, as opposed to infected, cells (*Figure 7C*). mAbs that induced ADNKA against infected cells also induced ADNKA against transfected cells (P008-014 and P054-044). However, of 12 antibodies that failed to activate ADNKA against infected cells, 5 induced potent ADNKA against transfected cells. Furthermore, in contrast to infected cells, the level of ADNKA induced by mAbs was extremely high against transfected cells, with many exceeding that of our control seropositive serum. Thus, despite a substantial ability to activate ADCC against cells transfected with spike protein, and high levels of antibody binding, spike was a relatively poor ADCC target in the context of natural infection when using mAbs.

## Following natural infection, ADNKA responses are dominated by non-spike antibodies

Although our data using monoclonal antibodies indicated that spike was a comparatively weak activator of ADNKA, the antibodies that mediate ADNKA against infected cells may represent a minor proportion of the overall anti-spike response; we simply may not have captured those antibodies in our panel. To directly address this concern, we specifically depleted anti-spike antibodies from the serum of naturally infected individuals, and assessed changes in activity. ELISA analysis confirmed the effective depletion of spike-specific antibodies (*Figure 8A*), which was accompanied by a significant loss of neutralisation activity (*Figure 8B*), across multiple donors (*Figure 8C*). However, there was no loss of ADNKA activity from any of these samples following spike-antibody depletion (*Figure 8D and E*), consistent with the results of our monoclonal antibody data. We conclude that spike-specific antibodies dominate the neutralising antibody response, but play a minor roles in the ADCC response following natural infection.

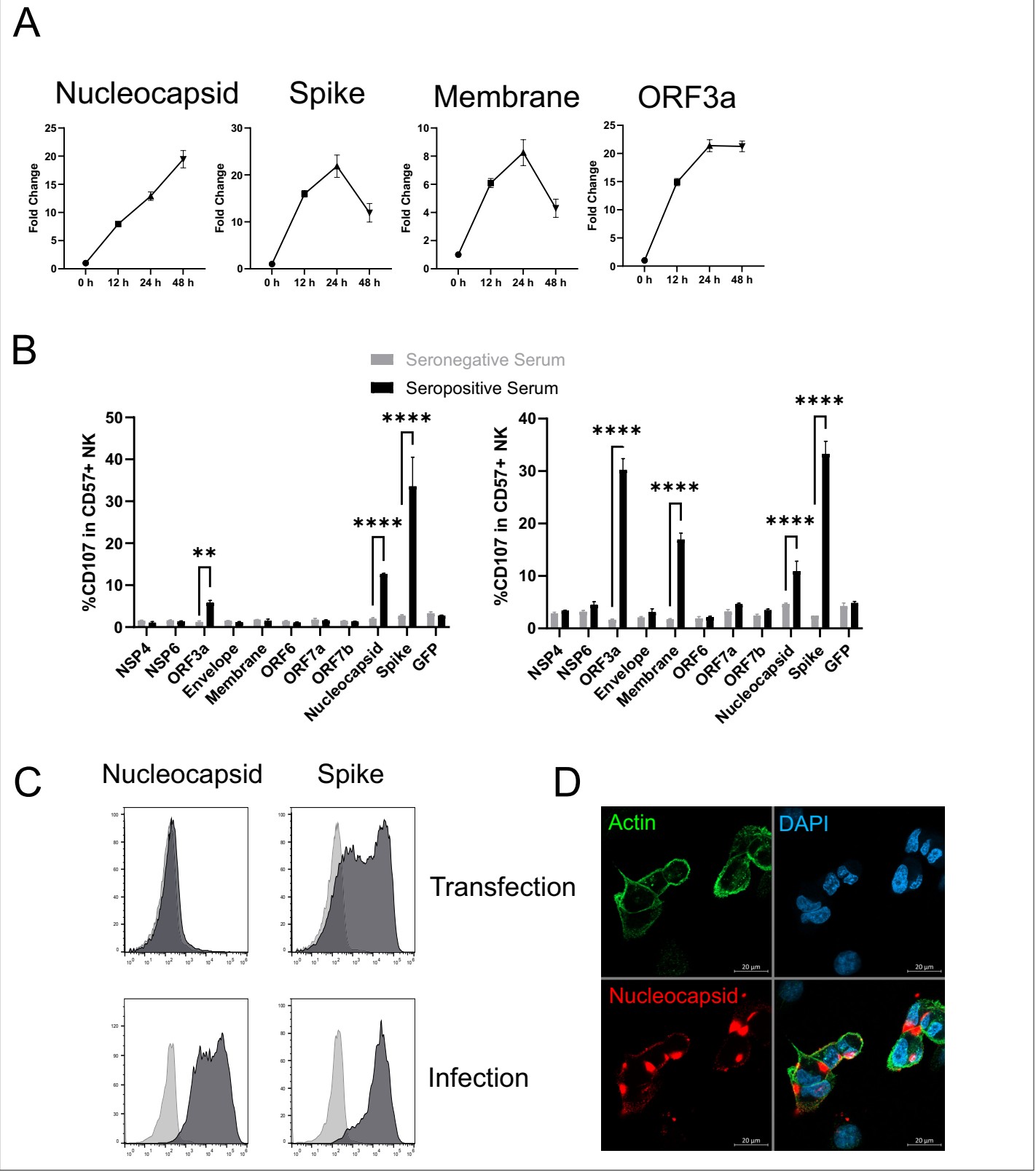

**Figure 6.** Multiple SARS-CoV-2 proteins mediate ADCC. (**A**) Plots of viral proteins detected in the PMP analysis. (**B**) 293T cells were transfected with plasmids expressing the indicated SARS-CoV-2 proteins for 48 hr, then mixed with PBMC in the presence of golgistop, CD107a antibody, and 1% serum from donors who were seronegative or seropositive for SARS-CoV-2. After 5 hr, cells were stained for CD3, CD56, CD57, and live/dead aqua, then analysed by flow cytometry. Assays were performed in technical triplicate. Data is shown from two donor serums. **p<0.01, ****p<0.0001, two-way

*Figure 6 continued on next page*

*Figure 6 continued*

ANOVA. (**C**) 293T cells were transfected with plasmids expressing SARS-CoV-2 nucleocapsid or spike (or empty vector; light grey) for 48 hr, or AAT cells were infected with SARS-CoV-2 (MOI = 5) (or mock infected; light grey) for 24 hr, then cells were detached with TrypLE, stained for nucleocapsid or spike, and analysed by flow cytometry. (**D**) AAT cells were infected with SARS-CoV-2 (MOI = 5) for 24 hr, then cells were fixed in paraformaldehyde and stained for the indicated targets. Images are shown as maximum intensity projections of Z-stacks.

The online version of this article includes the following figure supplement(s) for figure 6:

**Figure supplement 1.** Immunofluorescence staining of negative control cells from *Figure 6D*.

Since nucleocapsid, ORF3a, and membrane had activated ADNKA when expressed in isolation, we therefore investigated whether they also mediated ADNKA against virus infected cells. A nucleocapsid-specific human monoclonal antibody was capable of activating ADNKA against infected cells (*Figure 8F*) and, although not to the extent of polyclonal serum, was superior to our best anti-spike antibody. Furthermore, when we used the serums from which Spike antibodies had been depleted (*Figure 8D–E*), and also depleted anti-nucleocapsid antibodies, we saw a significant reduction in ADNKA activity (*Figure 8G–H*), although considerable activity remained. Reagents are not available to deplete antibodies targeting ORF3a or membrane, so we instead deleted ORF3a from the virus genome. In line with existing data, this resulted in a small reduction (~1 log) in virus titres (not shown). We therefore assessed levels of viral antigens on the cell surface following infection, which revealed that deletion of ORF3a resulted in a small reduction in levels of spike and nucleocapsid, but not membrane (*Figure 8I*). To ensure that these alterations did not affect readouts, we used serums that had been depleted for antibodies targeting both Spike and Nucleocapsid, and compared ADNKA responses against wildtype or ΔORF3a virus (*Figure 8J–K*). Deletion of ORF3a resulted in a dramatic reduction in ADNKA activity, although this was not reduced completely to the level of seronegative serum. We therefore conclude that nucleocapsid and ORF3a are targets for ADNKA activity against SARS-CoV-2 infected cells, with membrane potentially also contributing.

## Spike-specific antibodies following vaccination are weak mediators of ADNKA

Since non-spike antigens are required to efficiently prime ADCC following natural infection, we investigated individuals who have been vaccinated against spike, but not exposed to the virus – since they will only have spike-specific antibodies. As with serum from naturally infected individuals, ELISA demonstrated effective depletion of spike antibodies (*Figure 9A*), which was accompanied by a marked loss of virus neutralisation activity (*Figure 9B and C*). In functional NK assays, these sera only mediated weak ADNKA (*Figure 9D and E*) and, unlike sera from naturally infected individuals, ADNKA activity was abrogated following depletion of spike antibodies. When we compared responses following natural infection or vaccination, vaccine mediated ADNKA was significantly weaker than responses seen following natural infection, and was not boosted following the second dose of the vaccine (*Figure 9F*). This was in stark contrast with the neutralisation activity of those same serum samples, which were dramatically increased following the second dose, with activity comparable to patients with severe COVID-19 disease (*Figure 9G*).

Thus, anti-spike antibodies can mediate ADNKA, but this activity is significantly weaker than responses seen during natural infection, which targets a broader repertoire of viral antigens.

## Discussion

The host innate immune system plays a critical role in both early SARS-CoV-2 infection, and later COVID-19 disease. This is emphasised by the identification of multiple interferon antagonists encoded by SARS-CoV-2, and the increased disease severity in patients with genetic and acquired interferon deficiencies (*Lei et al., 2020*; *Sa Ribero et al., 2020*; *Miorin et al., 2020*; *Schroeder et al., 2021*). To determine whether the virus evades other components of the host immune system we took a proteomic approach to systematically analyse how SARS-CoV-2 infection manipulates the cell surface during the time course of SARS-CoV-2 infection. We find that SARS-CoV-2 remodelling of the plasma membrane leads to the downregulation of multiple cell surface immune ligands involved with the interferon response, cytokine function, and NK activation. This may enable SARS-CoV-2 to evade numerous effector arms of the host immune system. Of particular interest, NK activating ligands

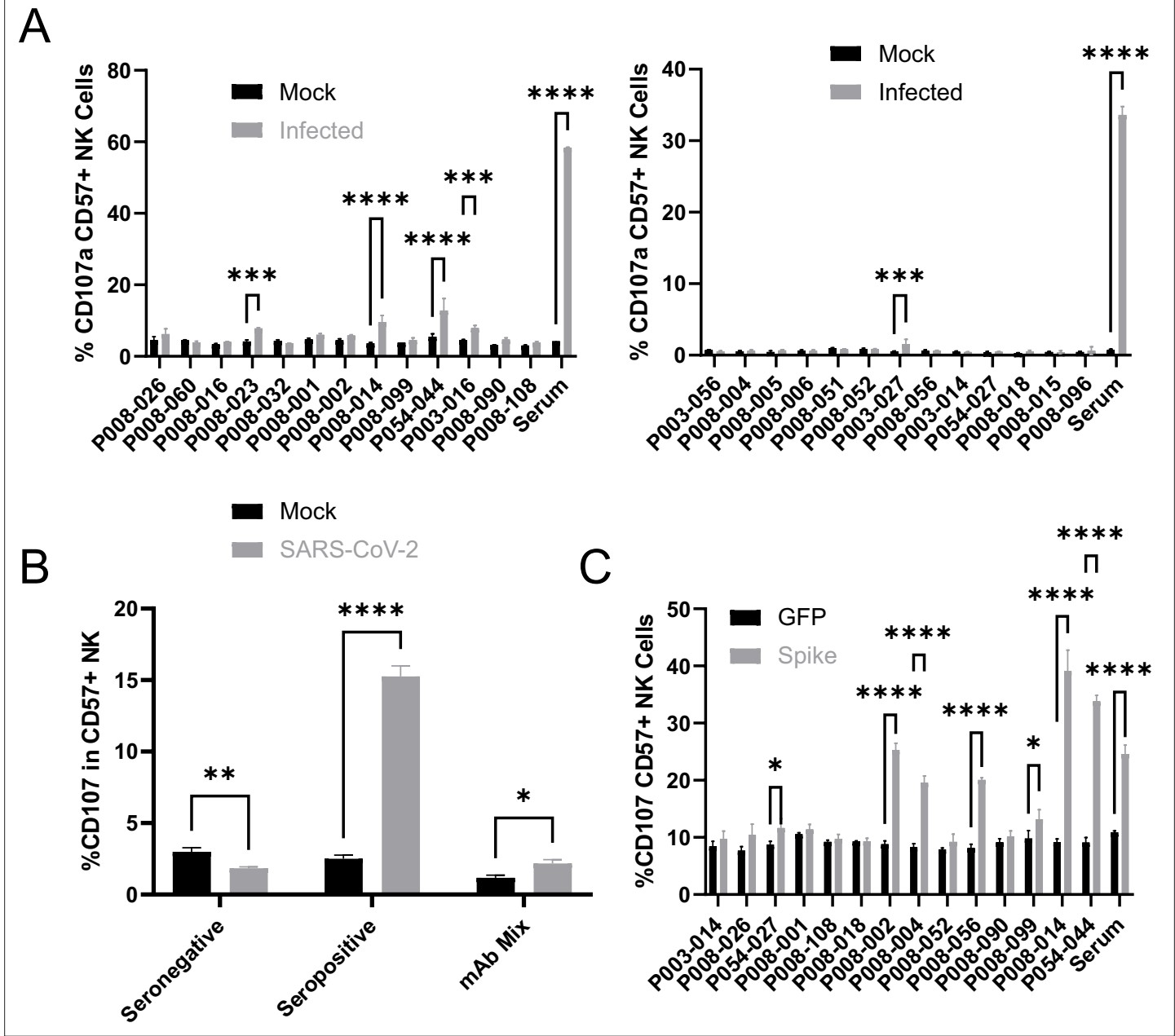

**Figure 7.** Monoclonal anti-spike antibodies bind infected cells strongly, but only weakly activate ADNKA. (**A–B**) AAT cells were either mock infected, or infected with SARS-CoV-2 (MOI = 5), or (**C**) 293T cells were transfected with an expression plasmid for spike. After 24 hr (**A–B**) or 48 hr (**C**), cells were detached with TrypLE and mixed with PBMC and the indicated antibodies, or a serum from a moderate case of COVID-19. Cells were incubated for 5 hr in the presence of golgistop, golgiplug, and CD107a antibody, before staining for CD3, CD56, CD57, and Live/Dead Aqua. Cells were gated on live CD57 +NK cells, and the percentage of cells positive for CD107a calculated. Assays were run in technical triplicate. Two-way ANOVA ***$p<0.001$, ****$p<0.0001$.

The online version of this article includes the following figure supplement(s) for figure 7:

**Figure supplement 1.** Anti-spike antibodies bind to SARS-CoV-2 infected cells.

**Figure supplement 2.** Monoclonal anti-spike antibodies only weakly activate ADNKA.

appear particularly susceptible to viral shutoff of host gene expression, and this results in inhibition of NK cell activation. This provides an explanation for how small RNA viruses may be able to target the NK response without encoding the large repertoire of specific NK-evasins seen in the large DNA viruses (*Patel et al., 2018*; *Berry et al., 2020*), and indicates that shutdown of host transcription

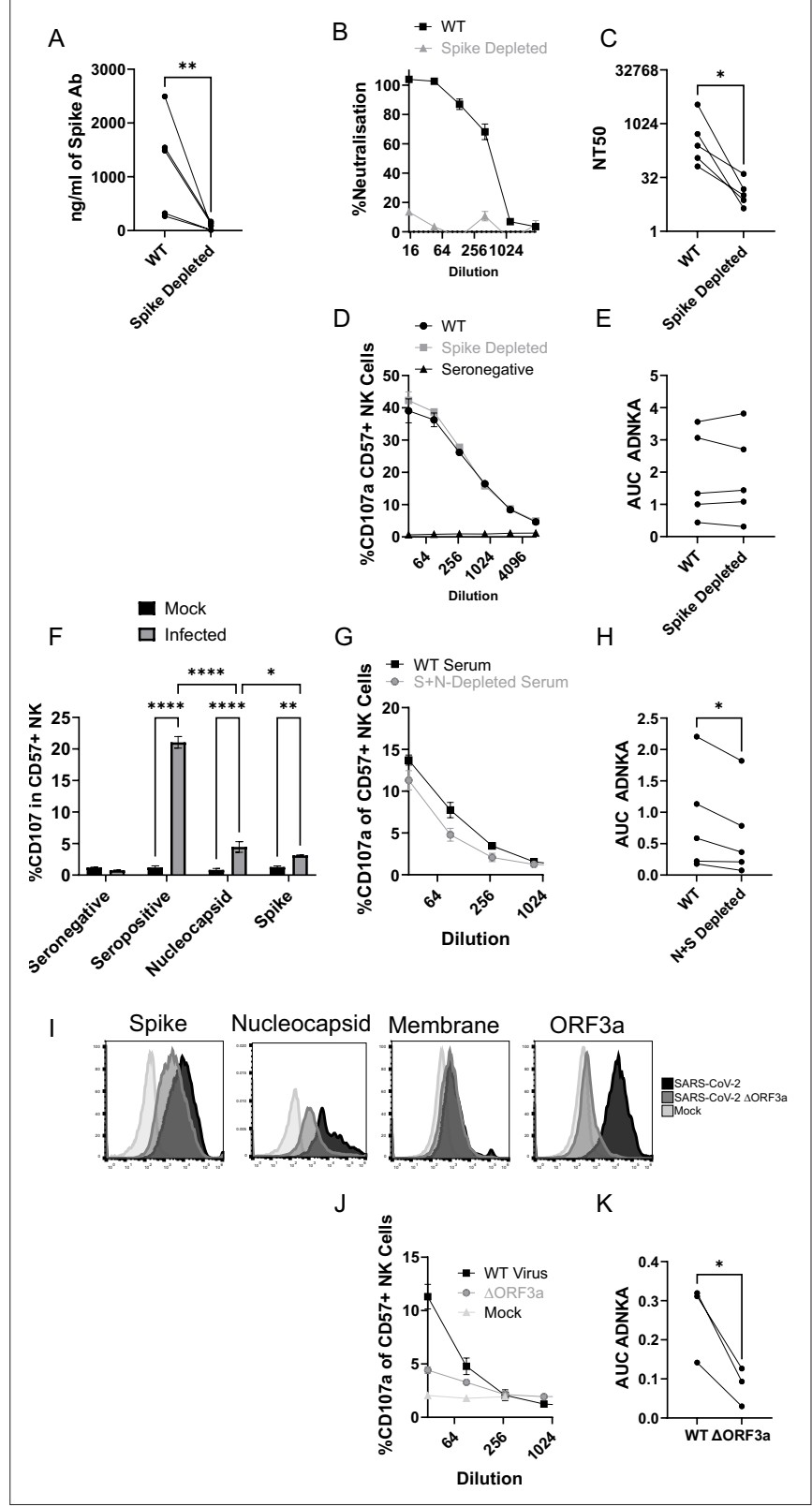

**Figure 8.** The ADNKA response is dominated by non-spike antibodies following natural infection. Serums from individuals naturally infected with SARS-CoV-2 were depleted of anti-spike antibodies using spike trimer protein (**A–E**) or both spike trimer and nucleocapsid (**G–H**) conjugated to magnetic beads. (**A**) ELISA for spike trimer was used to measure levels of antibodies before and after depletion. (**B, C**) the ability of the original, or anti-spike

*Figure 8 continued on next page*

*Figure 8 continued*

depleted, serums to neutralise the ability of SARS-CoV-2 to infect VeroE6 cells was determined across a range of concentrations, then NT50 values calculated. Example plots (**B**) and NT50 values for multiple donors (**C**) are shown. (**D–K**) AAT cells were either mock infected, or infected with SARS-CoV-2 (**D–H**), or with SARS-CoV-2 from which ORF3 had been deleted (**J, K**) for 24 h (MOI = 5), detached using TrypLE, then mixed with PBMC in the presence of golgistop, CD107a antibody, and serial dilutions of serum, or monoclonal antibody. After 5 hr, cells were stained for CD3, CD56, CD57, and live/dead aqua, then analysed by flow cytometry for the percentage of CD107a-positive CD57 +NK cells. AUC values were then calculated. Example plots (**D, G, J**) and AUC values for multiple donors (**E, H, K**) are shown. Kruskal–Wallis *p<0.05, **p<0.01, ****p<0.0001. (**I**) Twenty-four hr after infection, cells were detached and stained for the indicated proteins after analysis by flow cytometry.

and translation is likely to play a broader role in the ability of the virus to establish initial infection than previously appreciated. Later in disease, following the production of antibody, ADNKA leads to significant degranulation, and production of pro-inflammatory cytokines. Somewhat surprisingly, spike antibody is not responsible for the robust ADNKA which develops after natural infection, and only weak ADNKA is generated following spike vaccination.

Genetic variation in both viruses and their hosts affects the ability of viruses to modulate NK responses, and influence pathogenesis (*Patel et al., 2018*; *Vietzen et al., 2021a*). SARS-CoV-2 appears to have evolved to antagonise NK activation by targeting the expression levels of multiple activating NK ligands. Nectin-1 is a ligand for CD96/TACTILE (*Seth et al., 2007*), which activates human NK cells (*Holmes, 2019*), B7-H6 is a ligand for NKp30, while ULBP2 and MICA are ligands for NKG2D. NKG2D is ubiquitously expressed on γδ cells and CD8+, as well as some CD4+, T-cells in addition to NK cells (*Zingoni et al., 2018*). Targeting NKG2D ligands may therefore enable the virus to impact additional effector mechanisms beyond just NK cells. Amongst NKG2D ligands, MICA is particularly polymorphic (100+alleles), A549 cells express MICA*001/004 (*Seidel et al., 2015*), while the truncated MICA*008 is the most common allele worldwide. This diversity is largely driven by co-evolution of humans with virus infections, and reflects the broad repertoire of viral immune-evasins (*Zingoni et al., 2018*). MICA variation therefore has the potential to play a role in the extremely variable outcome of SARS-CoV-2 infection.

A recent study demonstrated that NK cells stimulated with IL-12 and IL-15 were capable of controlling SARS-CoV-2 following infection of VeroE6 or Calu3 cells (*Witkowski et al., 2021*). NK ligands are highly diverse between species, thus VeroE6 (i.e. African green monkey) cells cannot inform on the ability of human NK cells to control infection, while expression levels of NK ligands were not assessed on Calu3 cells, making it difficult to draw comparisons between the studies. Nevertheless, NK cell activity is heavily dependent on stimulation, and it is highly likely that strongly activating cytokines such as IL-12/IL-15 are capable of stimulating NK cells sufficiently to contribute to viral control despite viral evasion mechanisms.

In contrast to the effect on NK ligands, we observed minimal downregulation of MHC-I, implying that SARS-CoV-2 cannot prevent peptide presentation to antagonise adaptive cytotoxic cellular immunity. Nor do infected cells bind human IgG from seronegative donors, indicating that it does not encode Fc receptors as ADNKA decoys. These observations are consistent with SARS-CoV-2 acting as an acute 'hit and run' virus, which has replicated and transmitted before host adaptive immunity develops. A previous study has demonstrated that SARS-CoV-2 downregulates HLA-A2 via ORF8, which contrasts somewhat with our results (*Zhang et al., 2021*). This may indicate a HLA- or cell-type-specific effect, it will be interesting to determine whether coronaviruses evade innate and adaptive immunity in their original bat host, and whether this contributes to their persistence in that species.

In animal models of vaccination (*Gorman et al., 2021*) and monoclonal antibody administration (*Chan et al., 2020*; *Schäfer et al., 2021*; *Winkler et al., 2020*; *Suryadevara et al., 2021*; *Yamin et al., 2021*) against SARS-CoV-2, antibody-mediated activation of cellular immunity is a critical component of the protective immune response. However, inflammation dominates severe COVID-19 disease (*Horby et al., 2021b*; *Horby et al., 2021a*). The high levels of TNFα and IFNγ observed following ADNKA are likely to exacerbate disease under these circumstances. In support of this, TNFα and IFNγ correlate with severe disease (*Karki et al., 2021*), and critically ill patients have a higher proportion of afucosylated antibodies that may promote a stronger ADCC response (*Larsen et al., 2021*). Given that ADCC can potentially contribute to both protection and immunopathology, it is important to

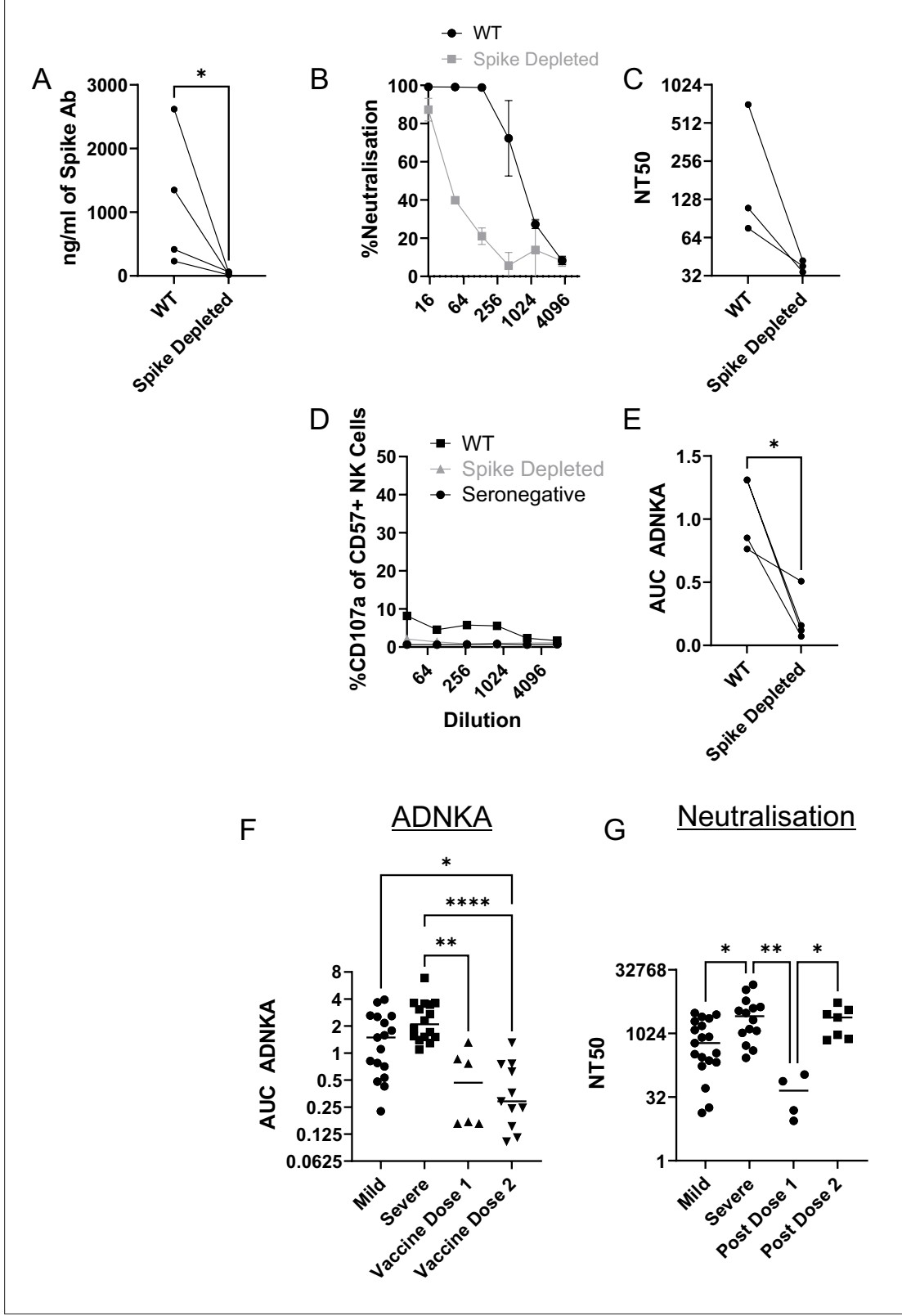

**Figure 9.** Vaccine responses focussed on spike are weak activators of ADNKA. Serums from donors that had been vaccinated against SARS-CoV-2 but were seronegative prior to vaccination were depleted of anti-spike antibodies using spike trimer protein conjugated to magnetic beads. (**A**) ELISA for spike trimer was used to measure levels of antibodies before and after depletion. (**B, C**) The ability of the original, or anti-spike depleted, serums to neutralise the ability of SARS-CoV-2 to infect VeroE6 cells was determined across a range of concentrations, then NT50 values calculated. Example

*Figure 9 continued on next page*

*Figure 9 continued*

plots (**B**) and NT50 values for multiple donors (**C**) are shown. (**D, E**) AAT cells were either mock infected, or infected with SARS-CoV-2 for 24 h (MOI = 5), detached using TrypLE, then mixed with PBMC in the presence of golgistop, CD107a antibody, and serial dilutions of serum. After 5 hr, cells were stained for CD3, CD56, CD57, and live/dead aqua, then analysed by flow cytometry for the percentage of CD107a positive CD57 +NK cells. AUC values were then calculated. All serums were additionally tested against mock infected cells to ensure no non-specific NK activation occurred (not shown). Example plots (**D**) and AUC values for multiple donors (**E**) are shown. Note that (**D**) is plotted on the same scale as *Figure 8D* to enable easy comparisons. Neutralisation (**F**) and ADNKA (**G**) values are compared for samples categorised according to disease or vaccination status. Kruskal–Wallis *p<0.05, **p<0.001.

understand what determines the magnitude and quality of the ADCC response. Viral entry glycoproteins are a common focus as mediators of ADCC. For example ADCC mediated by Env antibodies have been a major focus in HIV (*Forthal and Finzi, 2018*), and gB-specific antibodies have been studied in HCMV (*Nelson et al., 2018*). Although antibodies targeting these antigens can clearly bind infected cells, using similar approaches to the present study we recently showed that non-structural accessory proteins were more potent ADCC targets during HCMV infection (*Vlahava et al., 2021*). This conclusion is reinforced here, where the neutralising antibody response is dominated by spike, but spike antibodies are poor activators of ADNKA, with other viral antigens driving more efficient ADNKA following natural infection. In support of this data, monoclonal anti-spike antibodies were only effective at activating Fc-mediated control of infection in animal models following engineering to enhance activity (*Beaudoin-Bussières et al., 2021*). Numerous spike mAbs were capable of potently activating ADNKA against transfected cells, but failed to stimulate NK cells in the presence of infected cells; this may result from differential post-translational modifications of spike between these contexts, the fact that the reduction in activating NK ligands as a result of infection alters the threshold for activation through ADNKA, or the fact that viral CPE affects the formation of an immunological synapse (*Stanton et al., 2014*). Nevertheless, this underscores the importance of testing immune responses against live virus infected cells in order to assess efficacy. Importantly, the ability of monoclonal spike antibodies to bind to infected cells did not correlate with their ability to mediate ADNKA, implying that the epitope, position, or angle, at which the antibody binds to the cell surface is critical; it's notable that 4/5 anti-spike antibodies that activated ADNKA bound the NTD, suggesting that this site is structurally advantageous for Fc-mediated responses.

A similar phenomenon may underly the greater ability of non-spike antigens to generate a ADNKA response. Nucleocapsid, ORF3a, and membrane are all present on the infected cell surface, and can promote ADNKA if appropriate antibodies are present. In some studies, antibodies targeting ORF3a and membrane have been reported as present in only a subset of patients, while almost everyone developed antibodies targeting nucleocapsid (*Shrock et al., 2020*). However, other studies have shown that antibodies targeting ORF3a and membrane are much more common and found in most patients (*Heffron et al., 2021*). Our functional work is consistent with this latter data, and suggests that antibodies targeting all three proteins may be major targets for ADCC during natural SARS-CoV-2 infection, identifying a potential functional role for antibodies binding these immunodominant targets for the first time. Although it is surprising to observe nucleocapsid on the cell surface, Influenza nucleoprotein is also detected at the plasma membrane (*Yewdell et al., 1981*; *Fujimoto et al., 2016*; *Virelizier et al., 1977*), and antibodies targeting nucleoprotein can mediate ADCC (*Vanderven et al., 2016*; *Jegaskanda et al., 2017*; *Laidlaw et al., 2013*). Structural proteins involved in genome packaging may therefore represent a common target for ADCC across multiple virus families.

The identification of the antigens that drive ADCC is important to our understanding of the mechanisms that underlie pathogenesis and disease. A skewing of the spike:nucleocapsid antibody ratio is associated with severe disease (*Röltgen et al., 2020*; *de Campos Mata et al., 2020*; *Atyeo et al., 2020*), where ADCC-mediating nucleocapsid antibodies might be promoting inflammation. Both the breadth of the nucleocapsid antibody response, and the particular epitopes targeted, have also been correlated with disease severity (*Shrock et al., 2020*; *Voss et al., 2021*). These correlations now need to be followed up functionally, to determine how the epitopes bound by anti-nucleocapsid antibodies affect ADCC. Critically, the same surface-bound antibodies that activate NK cells are likely to promote inflammation by activating myeloid cells and complement (*Schulte-Schrepping et al., 2020*; *Szabo et al., 2021*; *Merad and Martin, 2020*), both of which are hallmarks of severe COVID-19.

Although Fc-dependent mechanisms of cellular immune activation could contribute to immunopathology via multiple mechanisms in severe disease, all patients with mild/asymptomatic disease also developed strong and persistent ADNKA responses. Furthermore, lineage defining mutations in the viral variants of concern B.1.1.7, B.1.351, and P.1, include non-synonymous mutations in membrane, ORF3a, and nucleocapsid, and these can result in reduced antibody binding (*Moore et al., 2021*; *Del Vecchio et al., 2021*). Virus variants may therefore evolve to evade ADCC responses as well as neutralising responses, supporting an important role for ADCC in protection. It is therefore significant that spike-vaccine generated antibody responses were poor ADNKA inducers when tested against infected cells. The addition of other viral proteins such as nucleocapsid to vaccines would engage a wider range of immune effector pathways. This approach might improve efficacy against both viral transmission and disease, resulting in vaccines that are more resistant to viral variants containing mutations that diminish antibody neutralisation (*Madhi et al., 2021*).

# Materials and methods

## Key resources table

| Reagent type (species) or resource | Designation | Source or reference | Identifiers | Additional information |
|---|---|---|---|---|
| Strain, Strain Background (SARS-CoV-2) | England 2 | Public Health England | Genome identical to NC_045512 | |
| Cell Line (Human) | A549 | ATCC | CCL-185 | |
| Cell Line (African Green Monkey) | VeroE6 | ATCC | CCL-81 | |
| Antibody | Mouse Monoclonal anti-CD107a-FITC | Biolegend | Cat No. 328,606 | Flow cytometry (1:100) |
| Antibody | Mouse monoclonal anti-CD56-BV605 | Biolegend | Cat No. 362,538 | Flow cytometry (1:100) |
| Antibody | Mouse monoclonal anti-CD3-PC7 | Biolegend | Cat No. 300,420 | Flow cytometry (1:100) |
| Antibody | Mouse monoclonal anti-CD57-APC | Biolegend | Cat No. 359,610 | Flow cytometry (1:100) |
| Antibody | Mouse monoclonal anti-B7-H6 | Biotechne | Cat No. MAB7144 | Flow cytometry (1:50) |
| Antibody | Mouse monoclonal anti-Nectin-1 | Biolegend | Cat No. 340,402 | Flow cytometry (1:200) |
| Antibody | Mouse monoclonal anti-MICA | BAMOMAB | Cat No. AMO1-100 | Flow cytometry (1:200) |
| Antibody | Mouse monoclonal anti-ULBP2 | BAMOMAB | Cat No. BUMO1 | Flow cytometry (1:200) |
| Antibody | Mouse monoclonal anti-Spike | Insight | Cat No. GTX632604 | Flow cytometry (1:500) |
| Antibody | Mouse monoclonal anti-Nucleocapsid | Stratech | Cat No. BSM-41411M | Flow Cytometry (1:300) |
| Antibody | Goat polyclonal anti-mouse IgG-AF647 | Thermo Fisher | Cat No. A-21235 | Flow Cytometry (1:500) |
| Antibody | Mouse monoclonal CD3-BV711 | Biolegend | Cat No. 344,837 | Flow cytometry (1:100) |
| Antibody | Mouse monoclonal CD57-PECy7 | Biolegend | Cat No. 359,623 | Flow cytometry (1:100) |
| Antibody | Mouse monoclonal TNFα–BV421 | Biolegend | Cat No. 502,931 | Flow cytometry (1:100) |
| Antibody | Mouse monoclonal IFNγ–APC | Biolegend | Cat No. 502,511 | Flow cytometry (1:100) |
| Antibody | Goat polyclonal Anti-mouse AF594 | Thermofisher | Cat No. A48288 | IFA (1:500) |
| Antibody | Human monoclonal anti-nucleocapsid | Acro | Cat No. NUN-S41 | Flow cytometry (10 µg/ml) |
| Other | Phalloidin-AF488 | Thermofisher | Cat No. A12379 | IFA (1:100) |
| Commercial assay, kit | Anti-spike | Acro | RAS-T025 | |
| Commercial assay, kit | Anti-Spike Magnetic beads | Acro | MBS-K015 | |
| Commercial assay, kit | Anti-nucleocapsid Magnetic Beads | Acro | MBS-K017 | |

## Cells and viruses

All cell lines tested negative for mycoplasma, and A549 were authenticated by Short tandem Repeat analysis. A549 were transduced with lentiviruses expressing human ACE2, and TMPRSS2 (AAT cells), as previously described (*Rihn et al., 2021*). The England2 strain of SARS-CoV-2 was obtained from Public Health England (PHE), and grown on VeroE6 cells. This strain has a genome

that is identical to the original Wuhan isolate. A virus lacking ORF3a, along with the parental virus, was a kind gift from Luis Martinez-Sobrido (Texas Biomedical Research Institute; *Silvas et al., 2021*). Virions were concentrated by pelleting through a 30% sucrose cushion to remove contaminating soluble proteins, and titrated by plaque assay on both VeroE6, AAT, and Caco2, as previously described (*Rihn et al., 2021*). All cell lines were grown in DMEM containing 10% FCS (Gibco), at 37 °C and in 5% $CO_2$. Throughout the study, multiple batches of virus were used, and no overt differences were noted between them. For assays, cells were plated out the day before, then infected at a multiplicity of infection (MOI) of 5, for 1 hr on a rocking platform. The inoculum was removed, and fresh DMEM containing 2% FCS was added, then cells were incubated until the assay. To confirm that phenotypes were dependent on active virus replication, virus was inactivated by heating to 56 °C for 30 min.

## Plasma membrane profiling

Cell surface proteins were labelled essentially as described (*Weekes et al., 2014*; *Weekes et al., 2012*). Briefly, cells were incubated in a solution containing sodium periodate, aniline and aminooxy biotin to label predominantly sialic acid containing glycans at the cell surface. Cell lysates were then enriched for labelled proteins using streptavidin-agarose beads. After extensive washing trypsin was added to liberate peptides of the enriched proteins. The resulting peptide pools were dried prior to labelling with TMT reagents.

## TMT labelling and clean-up

Samples were resuspended in 21 μL 100 mM TEAB pH 8.5. After allowing to come to room temperature, 0.2 mg TMT reagents (Thermo Fisher) were resuspended in 9 μL anhydrous acetonitrile (ACN) which was added to the respective samples and incubated at room temperature for 1 hr. A 3 μL aliquot of each sample was taken and pooled to check TMT labelling efficiency and equality of loading by LC-MS. After checking each sample was at least 98% TMT labelled total reporter ion intensities were used to normalise the pooling of the remaining samples such that the final pool was as close to a 1:1 ratio of total peptide content between samples as possible. This final pool was then dried in a vacuum centrifuge to evaporate the majority of ACN form labelling. The sample was acidified to a final 0.1% Trifluoracetic Acid (TFA) (~200 μL volume) and fluoroacetic acid (FA) was added until the SDC visibly precipitated. Four volumes of ethyl acetate were then added and the sample vortexed vigorously for 10 s. Sample was then centrifuged at 15,000 g for 5 min at RT to effect phase separation. A gel loading pipette tip was used to withdraw the lower (aqueous) phase to a fresh low adhesion microfuge tube. The sample was then partially dried in a vacuum centrifuge and brought up to a final volume of 1 mL with 0.1% TFA. FA was added until the pH was <2, confirmed by spotting onto pH paper. The sample was then cleaned up by SPE using a 50 mg tC18 SepPak cartridge (Waters). The cartridge was wetted with 1 mL 100% Methanol followed by 1 mL ACN, equilibrated with 1 mL 0.1% TFA and the sample loaded slowly. The sample was passed twice over the cartridge. The cartridge was washed 3 x with 1 mL 0.1% TFA before eluting sequentially with 250 μL 40% ACN, 70% ACN and 80% ACN and dried in a vacuum centrifuge.

## Basic pH reversed phase fractionation

TMT labelled samples were resuspended in 40 μL 200 mM Ammonium formate pH10 and transferred to a glass HPLC vial. BpH-RP fractionation was conducted on an Ultimate 3,000 UHPLC system (Thermo Scientific) equipped with a 2.1 mm ×15 cm, 1.7μ Kinetex EVO column (Phenomenex). Solvent A was 3% ACN, Solvent B was 100% ACN, solvent C was 200 mM ammonium formate (pH 10). Throughout the analysis solvent C was kept at a constant 10%. The flow rate was 500 μL/min and UV was monitored at 280 nm. Samples were loaded in 90% A for 10 min before a gradient elution of 0–10% B over 10 min (curve 3), 10–34% B over 21 min (curve 5), 34–50% B over 5 mins (curve 5) followed by a 10 min wash with 90% B. 15 s (100 μL) fractions were collected throughout the run. Fractions containing peptide (as determined by A280) were recombined across the gradient to preserve orthogonality with on-line low pH RP separation. For example, fractions 1, 25, 49, 73, 97 are combined and dried in a vacuum centrifuge and stored at –20 °C until LC-MS analysis. 24 Fractions were generated in this manner.

## Mass spectrometry

Samples were analysed on an Orbitrap Fusion instrument on-line with an Ultimate 3000 RSLC nano UHPLC system (Thermo Fisher). Samples were resuspended in 10 µL 5% DMSO/1% TFA and all sample was injected. Trapping solvent was 0.1% TFA, analytical solvent A was 0.1% FA, solvent B was ACN with 0.1% FA. Samples were loaded onto a trapping column (300 µm x 5 mm PepMap cartridge trap (Thermo Fisher)) at 10 µL/min for 5 min at 60 degrees. Samples were then separated on a 75 cm x 75 µm i.d. 2 µm particle size PepMap C18 column (Thermo Fisher) at 55 degrees. The gradient was 3–10% B over 10 min, 10–35% B over 155 min, 35–45% B over 9 min followed by a wash at 95% B for 5 min and requilibration at 3% B. Eluted peptides were introduced by electrospray to the MS by applying 2.1kV to a stainless-steel emitter (5 cm x 30 µm (PepSep)). During the gradient elution, mass spectra were acquired with the parameters detailed in *Figure 1—figure supplement 2* using Tune v3.3 and Xcalibur v4.3 (Thermo Fisher).

## Data processing

Data were processed with PeaksX+, v10.5 (Bioinformatic Solutions). Processing parameters are shown in detail in *Figure 1—figure supplement 3*. Briefly, raw files were searched iteratively in three rounds, with unmatched de novo spectra (at 1% PSM FDR) from the previous search used as the input for the next. The three iterations were as follows (1) Swissprot Human (27/03/2020)+common contaminants, (2) the same databases as search 1 but permitting semi-specific cleavage, (3) trEMBL Human (27/03/2020), with specific cleavage rules. Proteins were then quantified using the parameters outlined in *Figure 1—figure supplement 3*. Identified proteins and their abundances were output to .csv format and further subjected to statistical analysis. The mass spectrometry proteomics data have been deposited to the ProteomeXchange Consortium via the PRIDE partner repository (*Perez-Riverol et al., 2019*) with the dataset identifier PXD025000 and 10.6019/PXD025000. *Supplementary file 1* contains the analysed data.

## Statistical analysis and K-means clustering

Prior to statistical analysis, proteins were filtered for those quantified across all TMT reporter channels and with more than one unique peptide. Only proteins with a plasma membrane related GO:CC annotation as previously described (*Weekes et al., 2010*) were then carried forward for statistical analysis. All SARS-CoV-2 proteins quantified with more than one unique peptide were also included. Statistical tests were performed with the aov and p.adjust functions in the R base stats package (version 4.0.3) to calculate Benjamini-Hochberg corrected p-values for changes in protein abundance across the measured time-points.

Proteins with a p-value <0.05 and a maximum fold-change across the time-course of >1.5-fold were selected for k-means clustering. Mean intensities of each time-point were scaled using the scale function in base R and utilised to cluster proteins into five groups using the kmeans function in R base stats package.

## Functional annotation clustering

Assessment of enriched gene annotation terms in temporal cluster one was carried out using the Functional Annotation Clustering tool at DAVID (david.ncifcrf.gov) v6.8 (*Huang et al., 2009*), using the default clustering settings for medium stringency and the following libraries: Uniprot UP_Keyword, GOTERM:MF_ALL, BIOCARTA, KEGG_PATHWAY and REACTOME_PATHWAY. Output clusters were curated with representative names based on the enriched terms within. The background for enrichment was a list of proteins detected in the PMP dataset with more than one peptides and a plasma membrane associated GO:CC term. Functional annotation clusters with an enrichment score of >1.5 are shown, and any clusters in which no individual annotation term had a Benjamini-Hochberg correct p-value of <0.05 were excluded.

## Biochemical analysis of NK ligands

Replication-deficient adenovirus vectors (RAd) expressing B7-H6, MICA, and ULBP2, have been described previously (*Fielding et al., 2017*). To analyse glycosylation of these proteins, cells were infected with RAd-MICA and RAd-GFP at MOI = 100 and RAd-B7-H6 and RAd-ULBP2 at MOI = 500, then 48 hr later cells were either mock-infected or infected with SARS-CoV-2 MOI = 5 for a further

24 hr. For some experiments, lysates were digested overnight at 37 °C with EndoH or PNGaseF (NEB) according to manufacturer's instructions. In some experiments, MG132 (10 µM), Leupeptin (200 µM), or cycloheximide (CHX, 2.5 µg/ml) inhibitors were added 18 hr prior to harvest. Samples were then analysed by SDS-PAGE followed by western blotting for proteins of interest. Membranes were stained with primary antibodies (anti-MICA/B, BAMOMAB, BAMO1, 1:2000; anti-B7-H6, Abcam, ab121794, 1:2000; anti-ULBP-2, R&D Systems, AF1298, 1:2000; anti-GFP, FL, SC-8334, Santa Cruz Biotechnology, 1:2000; anti-Spike, Genetex, 1A9, 1:2000; anti-Actin, Sigma, 1:2000), followed by anti-mouse, anti-rabbit or anti-goat HRP-conjugated secondary antibody, then developed using Supersignal West Pico, and imaged in a Syngene XX6.

## Immunofluorescence

Cells were plated on coverslips, then infected with SARS-CoV-2. After 24 hr, cells were fixed with 4% paraformaldehyde, then stained with nucleocapsid mAb (1C7), followed by anti-mouse AF594, Phalloidin-AF488, and DAPI. Coverslips were washed, then mounted in prolong gold anti-fade (Thermofisher), before being imaged on a Zeiss LSM800 confocal microscope.

## Flow cytometry

Cells were dissociated using HyQtase, then stained with primary antibody for 30 min at 4 °C. Following washing, they were incubated with secondary antibody, again for 30 min at 4 °C. Cells were washed, fixed in 4% paraformaldehyde, and data collected on an Attune cytometer (Thermofisher). In one experiment (*Figure 1A*), cells were fixed and permeabilised (Cytofix/Cytoperm, BD) before staining. Antibodies used were against HLA-ABC (W632; AbD Serotec), NCR3LG1/B7-H6 (MAB7144, Biotechne R&D Systems), Nectin 1 (R1.302; Biolegend), MICA (AMO1-100; BAMOMAB), MICB (BMO2-100; BAMOMAB), ULBP2 (BUMO1; BAMOMAB), Spike (1A9; Insight), Nucleocapsid (1C7; Stratech), anti-mouse IgG AF647 (Thermofisher).

## ELISA

To measure levels of spike antibodies, an ELISA using the spike trimer (Acro Biosystems) was used according to manufacturers instructions. Each sample was measured in duplicate, and compared to a standard curve. A pre-pandemic serum was included in each assay to define the cut-off.

## Plasmids and transfections

Lentivirus plasmids encoding each SARS-CoV-2 ORF individually, with a C-terminal twin-strep tag, were obtained from Addgene, and have been validated for expression previously (*Gordon et al., 2020*). Plasmids were midiprepped (Nucleobond Xtra Midi; Machery-Nagel), and transfected into 293T cells using GeneJuice (Merck) according to manufacturers' instructions.

## NK activation assays

PBMCs from healthy donors were thawed from liquid N2 storage, rested overnight in RPMI supplemented with 10% FCS, and L-glutamine (2 mM). For assays investigating viral inhibition of NK activation, PBMC were stimulated overnight with IFN-α (1,000 U/ml) to provide a baseline level of activation against which inhibition could be seen. For ADNKA this stimulation is not required, and so cells were used unstimulated (*Vlahava et al., 2021*). To confirm that phenotypes were due to direct effects on NK cells, NK cells were purified by depletion, using the human NK cell purification kit (Miltenyi) according to manufacturer's instructions. Target cells were harvested using TrypLE Express (Gibco), preincubated for 30 min with the relevant antibody or serum preparations, then mixed with effectors at an effector:target (E:T) ratio of 10:1 (PBMC) or 1:1 (purified NK cells) in the presence of golgistop (0.7 µl/ml, BD Biosciences), Brefeldin-A (1:1000, Biolegend) and anti-CD107a–FITC (clone H4A3, BioLegend). Cells were incubated for 5 hr, washed in cold PBS, and stained with live/dead Fixable Aqua (Thermo Fisher Scientific), anti-CD3–PECy7 (clone UCHT1, BioLegend) or anti-CD3-BV711 (Clone SK7, Biolegend), anti-CD56–BV605 (clone 5.1H11, BioLegend), anti-CD57–APC (clone HNK-1, BioLegend) or anti-CD57-PECy7 (clone HNK-1, Biolegend), and anti-NKG2C–PE (clone 134591, R&D Systems). In some experiments, cells were also fixed/permeabilized using Cytofix/Cytoperm (BD Biosciences) and stained with anti-TNFα–BV421 (clone MAb11, BioLegend) and anti-IFNγ–APC (clone 4 S.B3, BioLegend). Data were acquired using an AttuneNxT (Thermo Fisher) and analyzed

with Attune NxT software or FlowJo software version 10 (Tree Star). Individual assays were run in technical triplicate. To ensure inter-assay variation did not affect results, a donor serum demonstrating moderate ADCC activity against SARS-CoV-2 was included as a positive standard in every assay, while a serum collected before 2020 was included as a negative control in every assay. Where sera were tested at a range of dilutions, the area under the curve (AUC) was calculated using Graphpad Prism 9. This value was then normalised to the AUC for the standard serum in that particular assay. In all experiments, serums were additionally tested against mock infected cells to control for non-specific activation of NK cells by serum components. An example gating strategy is shown in *Figure 4—figure supplement 1*, and involved gating on NK cells defined as CD56 positive, CD3 negative. This strategy will capture all 'classical' NK cells; however, it will miss the recently described CD56-negative NK cell population. Since these are only present at substantial levels in specific conditions such as chronic HIV infection, and older people with HCMV or EBV infection (*Müller-Durovic et al., 2019*), and have not been described to expand during SARS-CoV-2 infection, they are unlikely to play a major role in the NK response examined herein.

## Serums

Serums were collected from healthy donors vaccinated with the BNT162b2 vaccine, a minimum of 3 weeks after the first dose, or 1 week after the second. A pre-vaccine sample was taken from every donor, and an ELISA for SARS-CoV-2 RBD performed as previously described (*Amanat et al., 2020*), to determine pre-exposure to live SARS-CoV-2. Longitudinal serums from naturally infected individuals have been described previously, clinical characteristics are in *Supplementary file 3*; (*Seow et al., 2020*). Those labelled 'mild' had severity scores of 0, 1, 2 or 3, and 'severe' was 4 or 5, as described previously (*Seow et al., 2020*).

## Virus neutralisation assays

600PFU of SARS-CoV-2 was incubated with appropriate dilutions of serum, in duplicate, for 1 hr, at 37 °C. The mixes were then added to pre-plated VeroE6 cells for 48 hr. After this time, monolayers were fixed with 4% PFA, permeabilised for 15 min with 0.5% NP-40, then blocked for 1 hr in PBS containing 0.1% Tween (PBST) and 3% non-fat milk. Primary antibody (anti-nucleocapsid 1C7, Stratech, 1:500 dilution) was added in PBST containing 1% non-fast milk and incubated for 1 hr at room temperature. After washing in PBST, secondary antibody (anti-mouse IgG-HRP, Pierce, 1:3000 dilution) was added in PBST containing 1% non-fat milk and incubated for 1 hr. Monolayers were washed again, developed using Sigmafast OPD according to manufacturers' instructions, and read on a Clariostar Omega plate reader. Wells containing no virus, virus but no antibody, and a standardised serum displaying moderate activity were included as controls in every experiment. NT50 were calculated in Graphpad Prism 9.

## Antibody depletions

Anti-spike or anti-nucleocapsid antibody was depleted from sera using magnetic bead conjugated spike trimer protein or nucleocapsid protein (Acrobiosystems). Beads were resuspended in PBS +0.05% BSA, then 50 µl serum was mixed with 150 µl beads. Mixtures were incubated on a rotating mixer at 4 °C overnight. Serum diluted in buffer alone was used as a control. Magnetic beads were then removed using a 3D printed magnetic stand. All values given in assays are corrected for this initial fourfold dilution. Levels of anti-spike antibody were measured using a spike trimer ELISA (Acrobiosystems).

## Acknowledgements

This work was partly funded by the MRC/NIHR through the UK Coronavirus Immunology Consortium (CiC; MR/V028448/1). RS was supported by the MRC (MR/S00971X/1), Wellcome Trust (204870/Z/16/Z), and Ser Cymru. ECYW was funded by the MRC (MR/P001602/1, MR/V000489/1). PJL was funded by the Wellcome Trust through a Principal Research Fellowship (210688/Z/18/Z), the MRC (MR/V011561/1), the Addenbrooke's Charitable Trust and the NIHR Cambridge Biomedical Research Centre. KLD was supported by King's Together Rapid COVID-19 Call, Huo Family Foundation Award, and a Wellcome Trust Multi-User Equipment Grant 208354/Z/17/Z. CG was supported

by the MRC-KCL Doctoral Training Partnership in Biomedical Sciences (MR/N013700/1). BM was supported by an NIHR Academic Clinical Fellowship in Combined Infection Training

## Additional information

### Funding

| Funder | Grant reference number | Author |
|---|---|---|
| National Institute for Health Research | MR/V028448/1 | Ceri Alan Fielding<br>Pragati Sabberwal<br>Eddie CY Wang<br>Richard J Stanton |
| Medical Research Council | MR/S00971X/1 | Richard J Stanton |
| Wellcome Trust | 204870/Z/16/Z | Ceri Alan Fielding<br>Eddie CY Wang<br>Richard J Stanton |
| Ser Cymru | | Ceri Alan Fielding<br>Eddie CY Wang<br>Richard J Stanton |
| Medical Research Council | MR/P001602/1 | Eddie CY Wang |
| Medical Research Council | MR/V000489/1 | Richard J Stanton |
| Wellcome Trust | 210688/Z/18/Z | Paul J Lehner |
| Medical Research Council | MR/V011561/1 | Paul J Lehner |
| Wellcome Trust | 208354/Z/17/Z | Katie Doores |
| Medical Research Council | MR/N013700/1 | Carl Graham |

The funders had no role in study design, data collection and interpretation, or the decision to submit the work for publication. For the purpose of Open Access, the authors have applied a CC BY public copyright license to any Author Accepted Manuscript version arising from this submission.

### Author contributions

Ceri Alan Fielding, Richard J Stanton, Conceptualization, Data curation, Formal analysis, Funding acquisition, Investigation, Methodology, Project administration, Supervision, Validation, Visualization, Writing - original draft, Writing - review and editing; Pragati Sabberwal, Edward JD Greenwood, Thomas WM Crozier, Wioleta Zelek, Formal analysis, Investigation, Methodology; James C Williamson, Formal analysis, Investigation, Methodology, Writing - review and editing; Jeffrey Seow, Formal analysis, Investigation, Methodology, Resources; Carl Graham, Isabella Huettner, Methodology, Resources; Jonathan D Edgeworth, Katie Doores, Methodology, Resources, Writing - review and editing; David A Price, Blair Merrick, Sam J Wilson, Funding acquisition, Methodology, Resources, Writing - review and editing; Paul B Morgan, Funding acquisition, Resources, Writing - review and editing; Kristin Ladell, Funding acquisition, Investigation, Methodology, Resources; Matthias Eberl, Funding acquisition, Investigation, Methodology, Project administration, Resources; Ian R Humphreys, Funding acquisition, Methodology, Project administration, Resources; Paul J Lehner, Funding acquisition, Methodology, Resources, Supervision, Writing - review and editing; Eddie CY Wang, Conceptualization, Funding acquisition, Investigation, Methodology, Project administration, Resources, Supervision, Writing - review and editing

### Author ORCIDs

Ceri Alan Fielding http://orcid.org/0000-0002-5817-3153
Edward JD Greenwood http://orcid.org/0000-0002-5224-0263
David A Price http://orcid.org/0000-0001-9416-2737
Matthias Eberl http://orcid.org/0000-0002-9390-5348
Sam J Wilson http://orcid.org/0000-0002-6065-0895
Paul J Lehner http://orcid.org/0000-0001-9383-1054

Eddie CY Wang [iD]http://orcid.org/0000-0002-2243-4964
Richard J Stanton [iD]http://orcid.org/0000-0002-6799-1182

## Ethics

Human subjects: PBMC were extracted from apheresis cones obtained from the Welsh Blood Service (WBS) via an ad-hoc agreement or from blood samples from healthy volunteers and stored in liquid N2 until use. Use of healthy volunteer PBMC for this project, including those from WBS, was ethically approved by the Cardiff University School of Medicine Research Ethics Committee (SMREC) nos. 20/55 and 20/101. Recruitment of healthy volunteers after vaccination was covered by the Cardiff University School of Medicine Research Ethics Committee under reference no. 18/04.

## Decision letter and Author response

Decision letter https://doi.org/10.7554/eLife.74489.sa1
Author response https://doi.org/10.7554/eLife.74489.sa2

## Additional files

### Supplementary files

- Supplementary file 1. Processed Plasma Membrane Proteomics dataset.
- Supplementary file 2. List of interferon inducible genes identified in the filtered PM dataset.
- Supplementary file 3. Clinical characteristics of patients giving longitudinal serum samples.
- Supplementary file 4. Monoclonal Anti-spike Antibodies used, data taken from *Seow et al., 2020*.
- MDAR checklist

- Source data 1. Western Blot Source Data Information. Raw files for *Figure 3B* are provided as follows: Data 1 = MICA, Data 2 = Actin, Data 3 = Spike (all samples from RAd-MICA experiment). Data 4 = ULBP2, Data 5 = Actin, Data 6 = Spike (all samples from RAd-ULBP2 experiment). Data 7 = B7-H6, Data 8 = Actin, Data 9 = Spike (all samples from RAd-B7-H6 experiment). Raw files for *Figure 3C* are provided as follows: Data 1 = MICA, Data 2 = B7-H6, Data 3 = Actin, Data 4 = Spike Raw Files for *Figure 4A* are provided as follows: Data 1 = MICA, Data 2 = B7-H6, Data 3 = GFP, Data 4 = Actin

### Data availability

All data generated are presented within the manuscript, with the exception of proteomics data. This has been uploaded to PRIDE (identifier PXD025000).

The following dataset was generated:

| Author(s) | Year | Dataset title | Dataset URL | Database and Identifier |
|---|---|---|---|---|
| Fielding CA, Sabberwal P, Williamson JC, Greenwood EJD, Crozier TWM, Zelek W, Seow J, Graham C, Huettner I, Edgeworth JD, Price D, Morgan BP, Ladell K, Eberl M, Humphreys IR, Merrick B, Doores K, Wilson SJ, Lehner PJ, Wang ECY, Stanton RJ | 2021 | Plasma Membrane Profiling Of SARS-CoV-2 Infected Cells | https://www.ebi.ac.uk/pride/archive/projects/PXD025000 | PRIDE, PXD025000 |

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
