## [Editor Report]

By using a systematic proteomics approach, the authors demonstrated that SARS-CoV2 remodels the plasma membrane of infected human epithelial cells. Although the study indicates the manipulation of different immune response pathways, it seems that in the focus of viral immunoevasion are natural killer (NK) cells, which play a crucial role in controlling early viral infection. However, antibody-dependent NK cell activation was observed later in the disease process. These findings could have implications for the understanding of SARS-CoV-2 control by the immune system and vaccine development.

---

## [Decision Letter]

**Decision letter after peer review:**

Thank you for submitting your article "ADNKA overcomes SARS-CoV2-mediated NK cell inhibition through non-spike antibodies" for consideration by *eLife*. Your article has been reviewed by 3 peer reviewers, and the evaluation has been overseen by a Reviewing Editor and Betty Diamond as the Senior Editor. The following individual involved in review of your submission has agreed to reveal their identity: Domenico Mavilio (Reviewer #2).

Essential revisions:

All three reviewers concluded that the manuscript provides novel insights into the SARS CoV-2 escape from natural killer (NK) cells signaling by downregulating the expression of cellular ligands for activating NK receptors. The authors also showed that strong antibody-dependent NK cell activation (ADNKA) developed later in the disease process and that ADNKA was primarily driven by viral proteins other than Spike. This aspect might be relevant as the COVID-19 vaccine currently in use are based on spike protein. The reviewers agreed that your findings would be of interest to scientists in the field of NK cell biology and SARS CoV-2 pathogenesis. However, although enthusiastic with the bulk results, the reviewers pointed out several specific aspects of the work that need to be addressed experimentally and/or clarified in the revised manuscript.

*Reviewer #1:*

The study 'ADNKA overcomes SARS-CoV2-mediated NK cell inhibition through non-spike antibodies' by Fielding et al. deals with NK cell response to SARS-CoV-2 infected cells. First, by using mass spectrometry the authors show altered cell surface protein expression on SARS-CoV-2-infected lung epithelial cells and downregulation of activating NK ligands such as B7-H6, MICA, ULBP2, and Nectin1. This regulation of NK cell ligands resulted in impaired NK cell response (reduction in NK cell degranulation, IFNγ, and TNFα production) towards SARS-CoV-2 infected cells. Furthermore, they show strong NK cell activation in the presence of serum containing anti-SARS-CoV2 antibodies, and that such response was sustained for up to 6 months following infection. Based on the plasma membrane profiling and the use of transfectants expressing SARS-CoV-2 proteins, they identified Spike, Nucleocapsid, Membrane, and ORF3a as potential targets for ADCC. Depleting spike-specific antibodies from sera confirmed their dominant role in neutralization, but these antibodies had a minor role in ADNKA compared to antibodies targeting other proteins. In agreement with this observation, ADNKA was weaker following vaccination as compared to natural infection and was not boosted by the second dose of vaccine. Overall, this is an important and well-performed study. However, several questions remained unanswered, which required additional experiments.

1. Only one cell line was used to analyze the regulation of NK cell ligands upon SARS-CoV-2 infection. Are the same NK ligands regulated by SARS-CoV-2 in other permissive cell lines?

2. It was published recently that NK cells can regulate SARS-CoV-2 fitness in cell culture (Witkowski et al. 2021 Nature). This paper should be cited and discussed in the context of the presented results.

3. Antibodies to Spike, Nucleocapsid, Membrane, and ORF3a induced NK cell activation when transfectants expressing respective proteins were used. Depletion of Spike antibodies did not affect NK cell activation when co-incubated with SARS-CoV-2-infected cells, and the conclusion was drawn that non-spike antibodies mediate NK cell activation. However, formal proof for this claim is missing. The experiment should be performed with depletion of some non-spike antibodies (Nucleocapsid, Membrane, or ORF3a) from sera to prove that their elimination will impair NK cell activation. This would be especially important for antibodies targeting Nucleocapsid as they are present in most individuals infected with SARS-CoV-2.

4. Spike antibodies can induce ADNKA when Spike is expressed in the context of transfection. The major question is why these antibodies do not provide ADNKA in the context of infection? They can bind to the surface of infected cells, so something interferes with their function.

5. Panel of Spike-specific monoclonal antibodies was used to show that Spike-directed antibodies poorly activate NK cells when bound to infected cells. However, it is not clear if these antibodies can provide ADCC at all. Were they tested for ADCC when co-incubated with Spike-expressing transfectants?

*Reviewer #2:*

This paper is of high interest to a broad audience as it provides new knowledge on the role of NK cells in the control of COVID-19 disease progression and vaccine efficacy and design.

First, the authors identified a novel mechanism by which SARS-CoV2 antagonises innate immunity. By performing a proteomic analysis of SARS-CoV-2- infected lung epithelial cells, the authors demonstrated that SARS-CoV-2 infection induced a downregulation of activating NK ligands. in vitro functional assays demonstrated that this correlated with reduction in NK cell activation. Moreover, they showed that strong antibody-dependent NK cell activation (ADNKA) developed later in the disease process. In particular, it was shown that ADNKA is not primarily driven by spike antibody but was dominated by antibody to other viral proteins such as nucleocapsid, membrane and ORF3a. Since the anti-COVID-19 vaccine currently in use is based on spike protein, this last aspect could be highly relevant in vaccine design and improvement of vaccine efficacy.

Although the manuscript is well-structured and the data analysis is rigorous and justify the conclusions, some major points deserve to be better addressed or developed.

To further extend the knowledge provided on the role of NK cells in the context of SARS-CoV-2 infection, the authors should also investigate the function of the different NK subsets. In particular, also CD56-/CD16+ NK cells – which are not included in the present analysis – deserve to be deeply investigated as these cells are greatly expanded during viral infection (Di Vito C, et al. Front Immunol. 2019. Doi:10.3389/fimmu.2019.01812).

Moreover, as at 48h after infection the activating NK ligands are still downregulated whereas HLA molecules increased their expression, functional assay could be performed also with AAT cells infected for 48h to investigate also at this time point the NK activity.

Finally, since NK activation depends also on the expression levels of activating and inhibitory NK cell receptors, it could be interesting to assess whether SARS-CoV-2 infection can modulate receptor expression on NK cells obtained from COVID-19 patients. In particular, it would be interesting to investigate the expression of NKp30 (receptor of B7-H6) and NKG2D (receptor for ULBP2 and MICA). Moreover, as an increased expression of HLA molecules was observed in infected cells at 48h it would be interesting to assess also the expression on KIRs.

– In the present study, NK cells were identified in viable CD3-/CD56+ cells and then the authors focused their attention on CD57+ NK cells (CD3-/CD56+). Nevertheless, this gating strategy has some weakness since: (i) it does not allow a specific exclusion of B cells from the analysis (CD19 should be added in the panel); (ii) it not includes CD16. This last aspect is particularly relevant since the absence of CD16 in the panel used in functional assays precludes the possibility of investigating the function of the different NK subsets. In particular, CD56-/CD16+ NK cells which are greatly expanded during viral infection (Di Vito C, et al. Front Immunol. 2019. doi:10.3389/fimmu.2019.01812) cannot be assessed with the gating strategy used. Considering the role of CD56-/CD16+ NK cells in viral infection and the crucial involvement of CD16 in ADNKA, an additional set of experiments using an improved gating strategy that include at least CD16 should be performed to address these aspects.

– At 48h after infection, activating NK ligands are still downregulated whereas HLA molecules increase their expression. Since these last ones could be either activating or inhibitory, do the Authors assessed the NK activity also in assays performed co-culturing NK cells with AAT cells infected for 48h?

*Reviewer #3:*

Fielding et al. have studied the response of NK cells towards SARS-CoV-2 infected cells. In an unbiased proteomics approach the cell surface proteome was determined by plasma membrane profiling, applying extracellular aminooxy-biotinylation to selectively isolate and measure surface proteins. The authors found regulation of NK cell ligands: ULBP2, Nectin1, MICA and B7-H6. MHC-I proteins, which can act as inhibitory NK cell ligands, were not downregulated. Four viral proteins (S, M, N and ORF3a) were expressed on the plasma membrane of SARS-CoV-2 infected cells. Whereas the authors describe regulation of activating NK cell ligands rather briefly, they found compelling evidence of ADCC forwarded by antibody recognition of surface expressed SARS-CoV-2 antigens. Interestingly, this effect was not primarily conducted by anti-Spike antibodies, but antibodies directed against the other cell surface antigens. This demonstrates that antibodies directed against other proteins than Spike can induce an antiviral response through NK cell mediated ADCC.

Figure 3: MICA, ULBP2, B7-H6 and Nectin1 were found to be downregulated by SARS-CoV-2 infection. The authors suggest that Nectin1 specificity by SARS-CoV-2 could be a preserved function between an original bat virus and the pandemic virus studied here, since Nectin1 from human and a bat species is well conserved. However, the authors did not study at which level the observed regulation of Nectin1 takes place. It is well established that in SARS-CoV-2 infected cells the Nsp1 proteins blocks translational activity of ribosomes. Therefore, the regulation of NK cell ligands might not be protein specific, but the result of a more general mechanism executed e.g. by Nsp1. Therefore, the level (i.e. pre-transcriptional or pre- or post -translational) at which NK cell ligands are regulated should be determined.

Transcriptional analysis would also help to understand the potential role of these NK ligands for the SARS-CoV-2 antiviral response, since neither the non-treated nor the heat-inactivated virus did induce expression of NK cell ligands at any time.

Lastly, to this part, it is not clear which trigger is decisive for reduced NK cell activation upon co-culture with SARS-CoV-2 infected cells. Blocking antibodies should be included to demonstrate the specificity and relation between Figure 3A and 3B.

Figure 5A: Is there a difference between this figure and figure 1C other than the addition of N and M proteins; could these be merged to one figure?

Figure 5B: In this figure the strongest ADNKA responses are directed against Spike using serum from two individuals. This does not fit with the results shown in Figure 7. The authors should discuss this controversy.

A previous study (Hachim, Kavian et al., NatImm 2020) showed robust antibody responses against ORF8. One can guess that ORF8 was excluded from this analysis, because the gene might be disrupted in the strain used here (England2). However, this is not mentioned and no reference for the sequence is given. This information is important and should be included. The issue with ORF8 is also of importance for regulation of MHC-I, since in a previous study ORF8 was shown to block MHC-I by lysosomal degradation (Zhang et al., PNAS 2021).

Figure 5C. The finding that the N protein can be detected on the surface of SARS-CoV-2 infected cells is surprising. Has this been reported of before for SARS-CoV-2 or other corona viruses? Is N surface expression found on all infected cells or a specific population of cells (e.g. dying cells)? If N is distributed to the surface only in dying cells it questions the importance of this antigen in triggering ADCC.

Figure 6. The authors show that recombinant anti-Spike antibodies bind to SARS-CoV-2 infected cells as determined by flow cytometry, but do not induce ADNKA. This could be due to structural hindrance. Is there an association between antibody epitope recognition and induction of ADNKA? Furthermore, is it possible to further induce CD16 activation by combining anti-S antibodies with varying specificities in one sample to mimic the situation in serum?

Figure 7. In this figure it is elegantly demonstrated that non-spike SARS-CoV-2 specific antibodies prime cells for ADNKA. Since new SARS-CoV-2 variants escape neutralization by mutation of the Spike protein, it would be relevant to know whether the ability to induce ADNKA still remains equally effective by sera from individuals infected with previous variants

This study demonstrates the ability of non-Spike SARS-CoV-2 antibodies to induce ADCC. It is indeed surprising that depletion of anti-Spike antibodies has only a modest effect on ADCC. To underline the reduced role of Spike in ADNKA, controls suggested for Figure 6 and 7 should be included. Furthermore, if this is the first report on N protein expression on the cell surface, this point should be strengthened, e.g. by immunofluorescence microscopy.

Premature data in Figure 2-3 appear rather distracting here. In my opinion the manuscript would gain in clarity if the ADNKA part would be handled on its own.

[Editors' note: further revisions were suggested prior to acceptance, as described below.]

Thank you for resubmitting your work entitled "SARS-CoV-2 host-shutoff impacts innate NK cell functions, but antibody-dependent NK activity is strongly activated through non-spike antibodies" for further consideration by *eLife*. Your revised article has been evaluated by Betty Diamond (Senior Editor) and a Reviewing Editor.

The manuscript has been improved but there are some remaining issues that need to be addressed, as outlined below:

*Reviewer #1:*

The authors have addressed all of my concerns in the revised version of the manuscript. I would suggest accepting the manuscript for publication.

*Reviewer #2:*

In this revised version of the manuscript by Fielding and colleagues titled "SARS-CoV-2 host-shutoff impacts innate NK cell functions, but antibody-dependent NK activity is strongly activated through non-spike antibodies", the Authors remarkably improved the quality of their manuscript, providing new data and discussing critical aspects thus addressing the Reviewers' comments and improving the scientific significance of the paper.

*Reviewer #3:*

The revised version of the manuscript by Fielding et al. has been much improved. It now contains more detailed data on the regulation of NKG2D and Nkp30 ligands. The authors investigated the level of regulation and were able to exclude the degradation and retention of these proteins. Instead, Fielding et al., describe a selective regulation by Nsp14 and also Nsp1. This clearly strengthens the manuscript.

In general, the points raised previously have been approached and discussed appropriately.

I recommend publication of the manuscript after the following change to the manuscript have been made.

One important control is still missing in Figure 6C: a control with uninfected cells should be included.

---

## [Author Response]

Reviewer #1:The study 'ADNKA overcomes SARS-CoV2-mediated NK cell inhibition through non-spike antibodies' by Fielding et al. deals with NK cell response to SARS-CoV-2 infected cells. First, by using mass spectrometry the authors show altered cell surface protein expression on SARS-CoV-2-infected lung epithelial cells and downregulation of activating NK ligands such as B7-H6, MICA, ULBP2, and Nectin1. This regulation of NK cell ligands resulted in impaired NK cell response (reduction in NK cell degranulation, IFNγ, and TNFα production) towards SARS-CoV-2 infected cells. Furthermore, they show strong NK cell activation in the presence of serum containing anti-SARS-CoV2 antibodies, and that such response was sustained for up to 6 months following infection. Based on the plasma membrane profiling and the use of transfectants expressing SARS-CoV-2 proteins, they identified Spike, Nucleocapsid, Membrane, and ORF3a as potential targets for ADCC. Depleting spike-specific antibodies from sera confirmed their dominant role in neutralization, but these antibodies had a minor role in ADNKA compared to antibodies targeting other proteins. In agreement with this observation, ADNKA was weaker following vaccination as compared to natural infection and was not boosted by the second dose of vaccine. Overall, this is an important and well-performed study. However, several questions remained unanswered, which required additional experiments.1. Only one cell line was used to analyze the regulation of NK cell ligands upon SARS-CoV-2 infection. Are the same NK ligands regulated by SARS-CoV-2 in other permissive cell lines?

We have now investigated the use of Caco2 and Calu3, which are the only other permissive cell types in wide use for SARS-COV2 infection. We were unable to get reliable high MOI infection of Calu3, despite subcloning high ACE2 expressing cells. We have therefore analysed NK ligand downregulation in Caco2 cells. Levels of NK ligands were lower in these cells as compared to A549, nevertheless a similar downregulation was observed (new Figure S3). See also downregulation of NK ligands in 293T cells by viral ORFs that we added in response to reviewer 3’s comments.

2. It was published recently that NK cells can regulate SARS-CoV-2 fitness in cell culture (Witkowski et al. 2021 Nature). This paper should be cited and discussed in the context of the presented results.

We have now added discussion of this point: Recently a study demonstrated that NK cells stimulated with IL-12 and IL-15 were capable of controlling SARS-CoV-2 following infection of VeroE6 or Calu3 cells^62^. NK ligands are highly diverse between species, thus VeroE6 (i.e. African green monkey) cells cannot inform on the ability of human NK cells to control infection, while the expression levels of NK ligands were not assessed on Calu3 cells, making it difficult to draw comparisons between the studies. Nevertheless, NK cell activity is heavily dependent on stimulation, and it is highly likely that strongly activating cytokines such as IL12/IL-15 are capable of stimulating NK cells sufficiently to contribute to viral control despite viral evasion mechanisms.

3. Antibodies to Spike, Nucleocapsid, Membrane, and ORF3a induced NK cell activation when transfectants expressing respective proteins were used. Depletion of Spike antibodies did not affect NK cell activation when co-incubated with SARS-CoV-2-infected cells, and the conclusion was drawn that non-spike antibodies mediate NK cell activation. However, formal proof for this claim is missing. The experiment should be performed with depletion of some non-spike antibodies (Nucleocapsid, Membrane, or ORF3a) from sera to prove that their elimination will impair NK cell activation. This would be especially important for antibodies targeting Nucleocapsid as they are present in most individuals infected with SARS-CoV-2.

We have now depleted nucleocapsid antibodies, and shown that this impacts ADNKA. We have also used a monoclonal nucleocapsid antibody to show that this antigen is a ADNKA target. We do not have the reagents to deplete ORF3a or Membrane and cannot delete Membrane from the viral genome. However, we were able to delete ORF3a from the viral genome, and demonstrate that this results in a major reduction in ADNKA. This provides formal proof that both Nucleocapsid and ORF3a can act as ADNKA targets, and has been added to the manuscript (Figure 8).

4. Spike antibodies can induce ADNKA when Spike is expressed in the context of transfection. The major question is why these antibodies do not provide ADNKA in the context of infection? They can bind to the surface of infected cells, so something interferes with their function.

We agree that this is an important question, however addressing it is a substantial body of work in its own right, involving extensive structural and immunological work. It is therefore beyond the scope of this paper and does not change our major conclusion – that Spike mAbs are not the main mediators of ADNKA, and that all studies that have used Spike transfected cells to assess function should be re-interpreted in this light. We therefore state in the text ‘Numerous Spike mAbs were capable of potently activating ADNKA against transfected cells, but failed to stimulate NK cells in the presence of infected cells; this may result from differential post-translational modifications of Spike between these contexts, the fact that the reduction in activating NK ligands as a result of infection alters the threshold for activation through ADNKA, or the fact that viral CPE affects the formation of an immunological synapse^70^. Nevertheless, this underscores the importance of testing immune responses against live virus infected cells in order to assess efficacy’.

5. Panel of Spike-specific monoclonal antibodies was used to show that Spike-directed antibodies poorly activate NK cells when bound to infected cells. However, it is not clear if these antibodies can provide ADCC at all. Were they tested for ADCC when co-incubated with Spike-expressing transfectants?

We have now performed ADNKA assays using Spike transfected cells. This shows that antibodies that mediate ADNKA against infection also act against transfected cells. However numerous antibodies that do not act against infected cells, do act against transfected cells. This strengthens our conclusion that antibodies must be tested against virus, as opposed to transfection systems.

Reviewer #2:[…]– In the present study, NK cells were identified in viable CD3-/CD56+ cells and then the authors focused their attention on CD57+ NK cells (CD3-/CD56+). Nevertheless, this gating strategy has some weakness since: (i) it does not allow a specific exclusion of B cells from the analysis (CD19 should be added in the panel); (ii) it not includes CD16. This last aspect is particularly relevant since the absence of CD16 in the panel used in functional assays precludes the possibility of investigating the function of the different NK subsets. In particular, CD56-/CD16+ NK cells which are greatly expanded during viral infection (Di Vito C, et al. Front Immunol. 2019. doi:10.3389/fimmu.2019.01812) cannot be assessed with the gating strategy used. Considering the role of CD56-/CD16+ NK cells in viral infection and the crucial involvement of CD16 in ADNKA, an additional set of experiments using an improved gating strategy that include at least CD16 should be performed to address these aspects.

It is not possible to include a CD16 gate in functional assays, because following Fc engagement CD16 is cleaved from the cell surface in order to permit serial engagement. Thus, CD16 positive cells become CD16 negative following ADNKA. Furthermore, although the CD56-/CD16+ population has been demonstrated to be important in certain situations, this largely involves those with HIV and very elderly HCMV/EBV seropositive individuals. Since our donors do not fall into these groups, the proportion of CD56-/CD16+ cells will be too low to permit effective analysis. As a result, it is not possible to assess the role of this population in our study. We have therefore added to the methods section ‘This strategy will capture all ‘classical’ NK cells, however it will miss the recently described CD56 negative NK cell population. Since these are only present at substantial levels in specific conditions such as chronic HIV infection, and older people with HCMV or EBV infection^88^, and have not been described to expand during SARS-CoV2 infection, they are unlikely to play a major role in the NK response examined herein.’

– At 48h after infection, activating NK ligands are still downregulated whereas HLA molecules increase their expression. Since these last ones could be either activating or inhibitory, do the Authors assessed the NK activity also in assays performed co-culturing NK cells with AAT cells infected for 48h?

Comparable results were seen at both 24h and 48h post-infection, and we now state as such in the manuscript.

Reviewer #3:Fielding et al. have studied the response of NK cells towards SARS-CoV-2 infected cells. In an unbiased proteomics approach the cell surface proteome was determined by plasma membrane profiling, applying extracellular aminooxy-biotinylation to selectively isolate and measure surface proteins. The authors found regulation of NK cell ligands: ULBP2, Nectin1, MICA and B7-H6. MHC-I proteins, which can act as inhibitory NK cell ligands, were not downregulated. Four viral proteins (S, M, N and ORF3a) were expressed on the plasma membrane of SARS-CoV-2 infected cells. Whereas the authors describe regulation of activating NK cell ligands rather briefly, they found compelling evidence of ADCC forwarded by antibody recognition of surface expressed SARS-CoV-2 antigens. Interestingly, this effect was not primarily conducted by anti-Spike antibodies, but antibodies directed against the other cell surface antigens. This demonstrates that antibodies directed against other proteins than Spike can induce an antiviral response through NK cell mediated ADCC.Figure 3: MICA, ULBP2, B7-H6 and Nectin1 were found to be downregulated by SARS-CoV-2 infection. The authors suggest that Nectin1 specificity by SARS-CoV-2 could be a preserved function between an original bat virus and the pandemic virus studied here, since Nectin1 from human and a bat species is well conserved. However, the authors did not study at which level the observed regulation of Nectin1 takes place. It is well established that in SARS-CoV-2 infected cells the Nsp1 proteins blocks translational activity of ribosomes. Therefore, the regulation of NK cell ligands might not be protein specific, but the result of a more general mechanism executed e.g. by Nsp1. Therefore, the level (i.e. pre-transcriptional or pre- or post -translational) at which NK cell ligands are regulated should be determined.

We have now performed biochemical analysis to understand how SARS-CoV2 is manipulating these ligands (Figure 3-4). This revealed that there is likely a block at the stage of synthesis, as opposed to degradation or retention. Further screening of the entire SARS-CoV2 ORFeome identified that Nsp14 is likely the primary driver of this process, with some contribution from Nsp1. Given that these proteins target both transcription and translation of host genes, it seems likely that this is an example of the virus achieving inhibition of NK cell activation through non-specific targeting of host cell gene expression. This is extremely interesting, and explains how small RNA viruses are able to target the NK response without encoding a vast repertoire of specific NK modulators, as the large DNA viruses do. Given these results, we have removed the section on bat ligands from the results.

Transcriptional analysis would also help to understand the potential role of these NK ligands for the SARS-CoV-2 antiviral response, since neither the non-treated nor the heat-inactivated virus did induce expression of NK cell ligands at any time.

As described above, we have now performed extensive biochemical analysis, and screened all SARS-CoV-2 ORFs to identify viral inhibition of host gene expression as the likely mechanism by which this occurs.

Lastly, to this part, it is not clear which trigger is decisive for reduced NK cell activation upon co-culture with SARS-CoV-2 infected cells. Blocking antibodies should be included to demonstrate the specificity and relation between Figure 3A and 3B.

We have attempted to perform these experiments using antibodies to MICA, ULBP2, and B7H6 that have been shown to block NK activation; such blocking antibodies to Nectin1 have not, to our knowledge, been reported in the literature. However, it was not possible to block the activity of activating NK ligands in the presence of virus, since the ligand is already downregulated by the virus, leaving nothing to block. Nor does blocking work in the absence of virus infection; this is in line with our extensive experience analysing the NK response to other virus infections and in other cell types (e.g. 10.1371/journal.ppat.1004058, 10.7554/*eLife*.22206). We have consistently found that the only way to demonstrate the impact of an activating ligand in the context of virus infection is to delete the viral gene that inhibits it. When you do this, infection occurs in the presence of increased NK activating ligand that hyper-activates the NK cell response, and this can be blocked with blocking antibodies. However, with SARS-CoV-2, we cannot remove Nsp14 without inactivating the virus. At this point, it is therefore not technically possible to formally demonstrate this point, and have therefore been careful to avoid claiming anywhere that we have directly shown that these ligands are responsible for the effects seen.

Figure 5A: Is there a difference between this figure and figure 1C other than the addition of N and M proteins; could these be merged to one figure?

Although the data is the same, we would prefer to avoid introducing the presence of ORF1ab and Nucleocapsid in the earlier figure, because introducing them requires an extensive explanation of their significance on the cell surface. It is therefore easier for the text flow to keep the figures as they are originally.

Figure 5B: In this figure the strongest ADNKA responses are directed against Spike using serum from two individuals. This does not fit with the results shown in Figure 7. The authors should discuss this controversy.

We have now added an additional figure that directly demonstrates the differential impact of ADCC in the context of transfection vs infection (Figure 7C). The differential activity of antibodies against infection vs transfection may be due to glycosylation, viral CPE affecting synapse formation, or be a consequence of the viral manipulation of innate NK ligands that we describe. An extensive program of structural, virological and immunological work will be needed to investigate this, and other, hypotheses properly. At this stage our major conclusion is that other antigens are far more potent than Spike as ADNKA targets, and that immune responses need to be assessed against live virus infected cells. We therefore state

in the text: ‘this may result from differential post-translational modifications of Spike between these contexts, the fact that the reduction in activating NK ligands as a result of infection alters the threshold for activation through ADNKA, or the fact that viral CPE affects the formation of an immunological synapse^70^. Nevertheless, this underscores the importance of testing immune responses against live virus infected cells in order to assess efficacy.’

A previous study (Hachim, Kavian et al., NatImm 2020) showed robust antibody responses against ORF8. One can guess that ORF8 was excluded from this analysis, because the gene might be disrupted in the strain used here (England2). However, this is not mentioned and no reference for the sequence is given. This information is important and should be included. The issue with ORF8 is also of importance for regulation of MHC-I, since in a previous study ORF8 was shown to block MHC-I by lysosomal degradation (Zhang et al., PNAS 2021).

We have added to the methods ‘This strain has a genome that is identical to the original Wuhan isolate.’ ORF8 is mutated in Α, but not Wave1 viruses such as England 2. We do see a very minor downregulation of MHC-I, but the magnitude is so low that it is unlikely to be biologically significant. Whether this represents cell-type or HLA-type specific effects of ORF8 is currently unclear, therefore we state in the text ‘A previous study has demonstrated that SARS-CoV2 downregulates HLA-A2 via ORF8, which contrasts somewhat with our results^63^. This may indicate a HLA- or cell-type specific effect’. However, a full exploration of this phenomenon is tangential to the main results of our study, and beyond the scope of our current manuscript. Although ORF8 antibodies may be useful diagnostically, we do not have any evidence that ORF8 is found on the cell surface – it was not detected in our PMP, and does not have a signal peptide or transmembrane domain. Nor does the paper by Hachim et al. show surface expression. Therefore, it does not currently meet the criteria to be considered as a ADNKA target.

Figure 5C. The finding that the N protein can be detected on the surface of SARS-CoV-2 infected cells is surprising. Has this been reported of before for SARS-CoV-2 or other corona viruses? Is N surface expression found on all infected cells or a specific population of cells (e.g. dying cells)? If N is distributed to the surface only in dying cells it questions the importance of this antigen in triggering ADCC.

Our studies exclude dead cells, therefore this staining is found on live, productively infected cells. We agree that it is surprising, and a major result of our work. This has not been reported for other coronaviruses, but this may simply be because everyone has assumed that it is intracellular, rather than testing it empirically. We have now performed immunofluorescence to provide further evidence for the presence of N on the cell surface (Figure 6D). We also note that nucleoprotein of Flu is also found on the cell surface, and is a target for ADCC. Thus, although this phenomenon is novel for coronaviruses, it is not unique to viruses in general, and we make this point in the discussion.

Figure 6. The authors show that recombinant anti-Spike antibodies bind to SARS-CoV-2 infected cells as determined by flow cytometry, but do not induce ADNKA. This could be due to structural hindrance. Is there an association between antibody epitope recognition and induction of ADNKA? Furthermore, is it possible to further induce CD16 activation by combining anti-S antibodies with varying specificities in one sample to mimic the situation in serum?

It is noticeable that all those that induced ADNKA were targeted at the NTD, and we make this point in the discussion. We have also performed an experiment in which we combined the mAbs. ADNKA still fell far short of the levels of activation seen with serum.

Figure 7. In this figure it is elegantly demonstrated that non-spike SARS-CoV-2 specific antibodies prime cells for ADNKA. Since new SARS-CoV-2 variants escape neutralization by mutation of the Spike protein, it would be relevant to know whether the ability to induce ADNKA still remains equally effective by sera from individuals infected with previous variants

This is an extremely interesting point, which we are currently following up. However, it is a massive program of work, which is far beyond the scope of the current paper, which is already extensive. Our initial data using VOCs indicates that we will need to use reverse genetics systems to make hybrid viruses containing known distinct epitopes from different variants in order to dissect the specific impact of antigenic changes in different proteins of the different VOCs. We will also need to source serums from a large number of people infected with known variants, which has become more complicated as successive waves of different variants have passed through the population. It is therefore impractical to add this level of experimentation to the current manuscript, and is not required in order for us to draw the conclusions that we make.

This study demonstrates the ability of non-Spike SARS-CoV-2 antibodies to induce ADCC. It is indeed surprising that depletion of anti-Spike antibodies has only a modest effect on ADCC. To underline the reduced role of Spike in ADNKA, controls suggested for Figure 6 and 7 should be included. Furthermore, if this is the first report on N protein expression on the cell surface, this point should be strengthened, e.g. by immunofluorescence microscopy.

We are unclear which controls the reviewer is suggesting for Figure 6 and 7, however these sections have now been extended to demonstrate that a Nucleocapsid monoclonal can mediate ADNKA, that depletion of Nucleocapsid antibodies impacts ADNKA, that deletion of ORF3a impacts ADNKA, and that combining Spike MAbs is insufficient to recapitulate the effects of serum. As above, we have now also performed immunofluorescence to demonstrate the presence of Nucleocapsid on the cell surface.

Premature data in Figure 2-3 appear rather distracting here. In my opinion the manuscript would gain in clarity if the ADNKA part would be handled on its own.

We respectfully disagree, we would argue that it is not possible to fully understand the ability of NK cells to control SARS-CoV2 without understanding both the antigens that prime activation, and the ability of the virus to antagonise that activation through manipulation of activatory and inhibitory NK ligands on the cell surface. The modulation of innate ligands described in the early part of the paper may well be the reason that Spike monoclonals are significantly less effective against infected, as opposed to transfected, cells.

[Editors' note: further revisions were suggested prior to acceptance, as described below.]

The manuscript has been improved but there are some remaining issues that need to be addressed, as outlined below:Reviewer #3:The revised version of the manuscript by Fielding et al. has been much improved. It now contains more detailed data on the regulation of NKG2D and Nkp30 ligands. The authors investigated the level of regulation and were able to exclude the degradation and retention of these proteins. Instead, Fielding et al., describe a selective regulation by Nsp14 and also Nsp1. This clearly strengthens the manuscript.In general, the points raised previously have been approached and discussed appropriately.I recommend publication of the manuscript after the following changes to the manuscript have been made.One important control is still missing in Figure 6C: a control with uninfected cells should be included.

This control was included, however it was not explicitly described in the legend. This has now been corrected.